# Bayesian Persuasion Is a Bargaining Game

## Abstract

Bayesian persuasion studies how a sender with an informational advantage can persuade a receiver with a different motive to take actions that benefit the sender. This problem is previously formulated from an equilibrium perspective, where the sender is to choose a Bayes correlated equilibrium and the receiver is willing to respect the signaling scheme based on posterior beliefs. However, evidence in real-world scenarios and studies in farsighted receivers suggest otherwise: senders tend to be much more honest than the equilibrium. In this work, we show that Bayesian persuasion is reducible to a bargaining game. This reduction suggests that the receiver in Bayesian persuasion can be aware of the game structure and can develop an anti-exploitation strategy. This equalizes the power of commitment of the two parties and prevents the sender from taking the maximum possible payoff. Through experiments on large language models, we demonstrate the receiver's retaliatory strategies and the sender's compromise to that. More findings on the impact of the context and alignments further suggest that bargaining behavior emerges in persuasion tasks. The insights given by our results have potential implications on various scenarios to reduce exploitation, improve equality, and improve social welfare.

## 1 Introduction

Communication plays a pivotal role in human society, shaping interactions and influencing decision-making processes. The concept of "cheap talk," introduced by Crawford & Sobel (1982), underscores the importance of verbal communication in strategic contexts. McCloskey's assertion that "one quarter of GDP is persuasion" illustrates how integral persuasion is to economic activities (McCloskey & Klamer, 1995). Recent analyses by Antioch (2013) suggest that this figure has risen to 30%, highlighting the growing significance of persuasive communication in the modern economy. In this landscape, the idea of Bayesian persuasion, proposed by Kamenica & Gentzkow (2011), emerges as a crucial framework, involving the transmission of signals to influence a receiver's beliefs and actions. This framework finds a wide range of applications across various domains, including routing systems (Das et al., 2017; Kremer et al., 2014), law enforcement deployment (Hernández & Neeman, 2022; Lazear, 2006), and grading in schools (Boleslavsky & Cotton, 2015; Ostrovsky & Schwarz, 2010), as surveyed by Kamenica (2019).

A series of studies through the lens of Bayesian Correlated Equilibrium (BCE) demonstrates the high relevance of Bayesian persuasion. According to Bergemann & Morris (2013) and Bergemann & Morris (2016), the Bayesian persuasion task can be viewed as a scenario where the sender selects and commits an optimal BCE for itself (Bergemann & Morris, 2019). A signaling scheme qualifies as a BCE if it is incentive compatible (or obedient), meaning a rational receiver will follow the indications of this signaling scheme. As noted by Dughmi (2017), the commitment step is essential to Bayesian persuasion, otherwise the model will degenerate to the cheap talk model.

While the analysis of BCE is elegant, the sender's signaling scheme at equilibrium may be regarded as deceptive and exploitative from the common sense perspective. In fact, the senders in Bayesian persuasion are honestly dishonest (Best & Quigley, 2016). In human society, such sender might not be tolerated with partially lying in long-term interactions, let alone committing to such exploitative strategies. The experiments in Haran & Shalvi (2020) indicate "honest advice is more persuasive than highly informed advice" for humans. Priester & Petty (1995) suggests that cognitive misers are more likely to accept when the sender is perceived as relatively honest.

While the commitment assumption is motivated by long-term incentives, studies in farsighted players suggest different outcomes than the equilibrium. In the context of repeated games, Gan et al. (2022) demonstrate that persuading a far-sighted receiver, where players engage in an MDP, is NP-hard. They restrict their attention to consider an advice-myopic receiver who acts as if the sender will not be present in the future, and bases decisions solely on prior beliefs. They also introduce a threat-based signaling scheme for the sender, employing a grim-trigger strategy (Friedman, 1971) where the sender stops providing information if the receiver deviates. Extending this framework, Bernasconi et al. (2024) demonstrate that Markovian signaling schemes are suboptimal in this context, and propose a subset of history-dependent signaling schemes called promise-form. This approach encodes past interactions as honest promises about receiver's future rewards. These behaviors are evidence that bargain behavior may emerge in persuasion tasks.

In this work, we show that the receiver can extend its policy space for more complex action rules, which could force the sender to provide more information. In fact, because the receiver is capable of obtaining the expected values of both parties in the persuasion task, they could develop anti-exploitative action rules through this awareness of the game structure. Having the action rules deployed demonstrates the power of commitment, where the party that committed first will have the power to choose among the subgame perfect equilibria. Since the player who commits first has an advantage, the sender and receiver reach a consensus on their joint strategy pair to commit together before the game begins. This requires them to agree on the expected payoff induced by the strategy pair. Following this, we show that solving a Bayesian persuasion task needs the solution of a bargaining game. Therefore Bayesian persuasion is reducible to a bargaining game.

We conduct experiments with LLMs on several bargaining and persuasion games. The agents demonstrate surprisingly similar dynamics and outcomes in persuasion and bargaining. In particular, we observe the receiver's retaliation strategy for anti-exploitation and the sender's compromise to that. We also demonstrate that mathematically equivalent scenarios could result in different outcomes given different contexts. This could be explained by Bayesian theory of mind (Baker, 2012) where different prior (e.g. prior on honesty and retaliation) could impact the game. Our results give a thorough understanding of persuasion, both to its mathematical formulation and its real-life realization. It provides insights into how receivers could have anti-exploitation and counter-exploitation strategies, and has the potential to improve equality and social welfare in a variety of scenarios.

## 2 PRELIMINARIES

In this section, we give a brief introduction to Bayesian persuasion and bargaining. Because there is a large number of notations, we summarize them in the glossary in Appendix A.

### 2.1 BAYESIAN PERSUASION

Bayesian persuasion is an one-shot two-player game where a sender $i$ is trying to send signals $\sigma \in \Sigma$ to influence the actions $a \in \mathbb{A}$ of a receiver $j$. We initially focus on an imperfect and incomplete information game (as described in Procedure 1), for it being a general extension of Bayesian persuasion. Although the procedure is linear and contains no loops or branches, there are many details and assumptions to pay attention to.

A neutral environment will first samples a state $s \in \mathbb{S}$ according to a prior distribution $\mu_0 \in \Delta(\mathbb{S})$. State $s$ is observable only for the sender, and it influence both players' payoffs. So the receiver has interests in state $s$ without knowing it, which gives an opportunity to the sender to influence the behavior of the receiver. This refers to the sender's informational advantage. Then the environment samples a type[1] $\tau \sim q(\cdot \mid s)$. Types $\tau \in \mathbb{T}$ are the receiver's partial observations. It can be seen by both the sender and receiver, while in the multi-receiver setting, it cannot be seen by other receivers. The canonical Bayesian persuasion settings assume that the receiver has no private information, so $\mathbb{T} = \{0\}$. It means that the emission function $q$ maps all the state to the same type $\tau = 0$.

Next, the sender sends a signal $\sigma \in \Sigma$ to the receiver. Its signaling scheme is defined as $\tilde{\varphi} : \mathbb{S} \to \Delta(\Sigma)$, and the signaling scheme set is $\tilde{\Psi}$. Then, the receiver takes an action $a \in \mathbb{A}$ given the signal.

---

[1]It is also named "signal" in Bergemann & Morris (2016). But $\phi$ is determined by the environment. The sender can only pick $q$. To avoid ambiguity we use the term "type" here.

Its action rule is defined as $\tilde{\pi} : \Sigma \to \mathbb{A}$, and the action rule set is $\tilde{\Pi}$. The receiver's behavior is default to be Bayesian, which gives the task name "Bayesian persuasion". It implements its policy $\tilde{\pi}$ (Line 6 in Procedure 1) like: **(1)** It guesses a state given the sent signal according to the Bayesian decision theory Hart et al. (2000); **(2)** It chooses the best response to its guessed state. The second step is assumed to be optimal. The sender's influence on the receiver acts only on the first step.

The payoff functions of the sender and receiver are defined as $r^i : \mathbb{S} \times \mathbb{A} \to \mathbb{R}$ and $r^j : \mathbb{S} \times \mathbb{A} \to \mathbb{R}$, respectively. They do not generally share the same payoff function, so the task is mixed-motive at the most time. Any payoff instance is independent to the sender's strategy value $\sigma$, so for the sender to get better payoffs, it must leverage its informational advantage to manipulate the receiver to act towards its (the sender's) favor.

---

**Procedure 1:** Timing of the Imperfect and Incomplete Information Game

---

**Input:** Prior distribution $\mu_0$, state set $\mathbb{S}$, the receiver's action set $\mathbb{A}$, reward functions $r^i$ and $r^j$

1 Given $\mu_0$, the sender decides a signaling scheme $\tilde{\varphi}$ ;            // Commitment (1)
2 Sender commits to $\tilde{\varphi}$ to the receiver ;            // Commitment (2)
3 Environment samples a state $s \sim \mu_0(\cdot)$ ;
4 Environment samples a type $\tau \sim q(\cdot \mid s)$ ;       // Dummy in Bayesian persuasion
5 The sender recommends an action $\sigma \sim \phi(\cdot \mid s, \tau)$ where $\sigma \in \mathbb{A}$ ; // Commitment (3) & Revelation
6 The receiver chooses an action $a \sim \tilde{\pi}(\cdot \mid \sigma)$ ;         // Default: Bayesian best response
7 The sender and receiver get rewards of $r^i(s, a)$ and $r^j(s, a)$ respectively ;

---

As stated, the receiver needs to make a Bayesian best response, which requires knowing the sender's signaling scheme $\tilde{\varphi}$. This is allowed in Bayesian persuasion and is referred to as **the commitment assumption**. This is a signature key assumption, which is indispensable in Bayesian persuasion problems; without this assumption, Bayesian persuasion would degenerate into cheap talk (Crawford & Sobel, 1982; Kamenica & Gentzkow, 2011).

**Assumption 2.1** (Commitment). *The sender commits to a signaling scheme $\tilde{\varphi}$, which encompasses the following three key aspects: (1) The sender will decide on a $\tilde{\varphi}$ before the game starts, (2) The sender will honestly inform the receiver of this $\tilde{\varphi}$, and (3) during the game, the sender will actually samples signals $\sigma \in \Sigma$ according to the committed $\tilde{\varphi}$.*

The reasoning behind this assumption is to simplify the analysis of the problem. Its justification is that the sender and receiver will interact over the long term, and considering its own reputation, the sender will not lie about the committed signaling scheme.

We can further simplify the problem without loss of generality by assuming that the sender "sending a signal" is equivalent to "recommending an action" for the receiver to take. And this is known as an analysis similar to the revelation principle, proposed and proved in Kamenica & Gentzkow (2011).

**Proposition 2.2** (A Variant of Revelation Principle [2], Paraphrased from Dughmi (2017)). *Assuming that the signaling set is equal to the action set ($\Sigma = \mathbb{A}$) does not affect the optimality of the sender's signaling scheme $\tilde{\varphi}$. This implies that every signaling scheme is equivalent to one that recommends actions.*

Let $\mathbb{V}^i$ and $\mathbb{V}^j$ denote the visibility set of the sender and receiver respectively. Let $\mathbb{V}^{i+j}$ denote the common knowledge set (known by the players, with each aware that others know it, ad infinitum). Then $\mathbb{V}^i \setminus \mathbb{V}^j = \{s\}$ and $i, j, \mu_0, \mathbb{S}, \mathbb{A}, r^j, \tilde{\varphi}, \sigma, a \in \mathbb{V}^{i+j}$. The visibility of $r^i$ is not specified in previous work. Here we assume $r^i \in \mathbb{V}^{i+j}$.

To further characterize the task so that the solutions to become conceivable, we restrict our attention to a subset of $\tilde{\Phi}$ whose elements are *signaling schemes that the receiver would respect*. This can be clearly introduced by the concept of **incentive compatibility**.

---

[2]We will refer to this as the revelation principle in the following content since there will be no ambiguity here. And we keep using the notation $\sigma$ to denote an action sent by the sender, using $a$ to denote an action actually taken by the receiver.

**Definition 2.3** (Incentive Compatibility in Bayesian Persuasion, Single-Receiver Version of Bergemann & Morris (2016)). *The sender's recommendation $\phi$ is incentive compatible for the receiver if for each $\tau \in \mathbb{T}$ and $a \in \mathbb{A}$,*

$$\sum_s \mu_0(s) \cdot q(\tau \mid s) \cdot \phi(a \mid s, \tau) \cdot r^j(s, a) \geq \sum_s \mu_0(s) \cdot q(\tau \mid s) \cdot \phi(a \mid s, \tau) \cdot r^j(s, a'), \quad (1)$$

*for all $a' \in \mathbb{A}$.*

Every signal (recommended action) will induce a posterior belief of the receiver, who will then choose the Bayesian best response to it. If the receiver's best response is exactly the action recommended by the sender, then the receiver is considered persuaded by the sender. In this case, the receiver has no incentives to deviate from the sender's recommendation, i.e., $\tilde{\pi}(a \mid \sigma = a) = 1$, and thus can be omitted in this canonical setting. In the origin paper Bergemann & Morris (2016), it is named as the "obedience". **Bayes correlated equilibria (BCE)** is defined as the set of obedient signaling schemes.

Since we assume $\mathbb{T} = \{0\}$ in this work, $\phi(a \mid s, \tau) \equiv \tilde{\varphi}(\sigma = a \mid s)$, and the inequality can be simplified as $\sum_s \mu_0(s) \cdot \tilde{\varphi}(a \mid s) \cdot \left[ r^j(s, a) - r^j(s, a') \right] \geq 0$. This is indeed the formulation in the survey Dughmi (2017).

Having these, the sender faces an optimization problem in the beginning of the game (Line 1 in Procedure 1):

$$\max_\varphi \quad \mathbb{E}_{\mu_0, \tilde{\varphi}, \tilde{\pi}} \left[ r^i(s, a) \right] := \sum_s \mu_0(s) \sum_a \tilde{\varphi}(\sigma = a \mid s) \cdot r^i(s, a)$$

$$\text{s.t.} \quad \sum_s \mu_0(s) \cdot \tilde{\varphi}(\sigma = a \mid s) \cdot \left[ r^j(s, a) - r^j(s, a') \right] \geq 0, \quad \forall a, a' \in \mathbb{A}. \tag{2}$$

That is, the sender is to select an incentive compatible signaling scheme for the receiver, to maximize its (the sender's) own expected payoffs. We refer to the canonical Bayesian persuasion as the formulation in Dughmi (2017). Concrete examples of persuasion are provided in Section 4.1.

**Definition 2.4** (Bayesian Persuasion). *Under Assumption 2.1 and Proposition 2.2, a Bayesian persuasion task is defined as $\mathcal{BP} := (\mathbb{S}, \mu_0, \mathbb{A}, r^i, r^j)$, where a sender tries to persuade a receiver according to Procedure 1.*

## 2.2 BARGAINING GAME

The bargaining game, initially introduced by Nash et al. (1950); Nash (1953), serves as a foundational model in the study of strategic negotiation. In this work, we follow the definition in Maschler et al. (2013).

**Definition 2.5** (Bargaining Game (Maschler et al., 2013)). *A two-player bargaining game is defined as $\mathcal{BG} := (\mathbb{Y}, \boldsymbol{d})$, where:*

- *$\mathbb{Y} \subseteq \mathbb{R}^2$ represents a nonempty and compact set known as the **feasibility set** (or agreement set), which is the set of all possible agreements.*

- *$\boldsymbol{d} = (d_i, d_j) \in \mathbb{R}^2$ denotes the **disagreement point**.*

- *There exists at least one agreement $y = (y_i, y_j) \in \mathbb{Y}$ such that $y_i > d_i$ and $y_j > d_j$.*

A critical element of the bargaining game is the presence of a disagreement point, which fundamentally enables each player to unilaterally refuse cooperation. Concrete examples of bargaining are provided in Section 4.1.

A classic perspective on solutions is axiomatic models, which propose certain axioms as necessary and sufficient conditions for a bargaining solution. Notable bargaining solutions include the Nash bargaining solution (Nash et al., 1950), the Kalai–Smorodinsky bargaining solution (Kalai & Smorodinsky, 1975), and the proportional bargaining solution (Kalai, 1977). In this work, we follow the definition of bargaining solution in (Maschler et al., 2013).

**Definition 2.6** (Solution Concept of Bargaining Game (Maschler et al., 2013)). *A solution concept of bargaining games is defined as a function $\omega_{\mathcal{BG}}$ which maps every bargaining game $(\mathbb{Y}, \boldsymbol{d})$ to an agreement $\boldsymbol{y} \in \mathbb{Y}$.*

## 3 REDUCTION FROM BAYESIAN PERSUASION TO BARGAINING GAMES

In this section, we show that Bayesian persuasion task is reducible to a bargaining game. The receiver has sufficient knowledge to calculate the game structure, and there always exists a disagreement point in Bayesian persuasion where the receiver can ignore the sender. At this disagreement point, the receiver can unilaterally set both players' payoffs to default values, which nullifies the sender's informational advantage. Thus the receiver has the ability to make a commitment before the game starts, just like the sender. From this intuition, Bayesian persuasion is reducible to a bargaining game. This reduction suggests that the receiver in Bayesian persuasion may have retaliatory strategies, and it opens up possibilities for further discussions in behavioral economics.

Our understanding of the existing related work that employs bargaining strategies in Bayesian persuasion is discussed in AppendixB.2.

### 3.1 THE RECEIVER'S AWARENESS OF THE GAME STRUCTURE

We start with recalling how the receiver will act before they reach a consensus, i.e., $\tilde{\pi}(a \mid \sigma = a)$ does not have to be 1 and thus cannot be simplified.

Since $\mu_0, \tilde{\varphi} \in \mathbb{V}^{i+j}$, given a $\tilde{\pi}$, the Bayesian receiver is able to calculate its ground truth expected payoff as $\sum_s \mu_0(s) \sum_\sigma \tilde{\varphi}(\sigma \mid s) \sum_{s'} \mu(s' \mid \sigma) \sum_a \tilde{\pi}(a \mid s') \cdot r^j(s, a)$, where $\mu(s \mid \sigma) = (\mu_0(s) \cdot \tilde{\varphi}(\sigma \mid s))/(\sum_{s'} \mu_0(s') \cdot \tilde{\varphi}(\sigma \mid s'))$ is the posterior belief according to Bayes' rule, and $\tilde{\pi}$ optimizes the receiver's payoff from its posterior view. Or more generally for an arbitrary decision rule $\tilde{\pi}$, the receiver's ground true expected payoff is

$$R^j(\mu_0, \tilde{\varphi}, \tilde{\pi}, r^j) := \sum_s \mu_0(s) \sum_\sigma \tilde{\varphi}(\sigma \mid s) \sum_a \tilde{\pi}(a \mid \sigma) \cdot r^j(s, a). \tag{3}$$

Because the receiver can see the sender's reward function ($r^i \in \mathbb{V}^{i+j}$), the receiver can also calculate the sender's expected payoff, by replacing $r^j(s, a)$ in Equation (3) with $r^i(s, a)$.

Define the **game structure** for every task $\mathcal{BP}$ as a mapping $\psi : \tilde{\tilde{\Phi}} \times \tilde{\tilde{\Pi}} \to \{\boldsymbol{R}\}$, where $\{\boldsymbol{R}\} := \{(R^i, R^j)\}$ is the set of all possible expected payoff pairs in $\mathcal{BP}$. The game structure is determined by the nature of the parameters and the intrinsic setting of Bayesian persuasion. It therefore can be reasoned out if the task parameters are specified. We can now conclude that $\psi \in \mathbb{V}^{i+j}$, since it can be reasoned out by both players using other common knowledge. Let $\mathbb{\Psi}$ denote all the possible game structure mappings for different $\mathcal{BP}$ tasks.

Recall that, given a committed signaling scheme $\tilde{\varphi}$ and a signal $\sigma$, the receiver will first calculate its posterior belief, guess a state $s'$ according to Bayesian decision theory, and then choose its best response to this guess. We call this default Bayesian best response as the action rule $\tilde{\pi}_1 := \text{BestResponseTo}(\mathcal{BP}, \tilde{\varphi}, \sigma)$. The corresponding expected payoff pairs is denoted as $\boldsymbol{R}_1 = (R_1^i, R_1^j) = \psi(\tilde{\varphi}, \tilde{\pi}_1)$.

### 3.2 EXISTENCE OF THE DISAGREEMENT POINT IN PERSUASION

In all cases, the receiver always has the choice to ignore the sender's signaling scheme and signals. It can calculate its best response solely on its prior knowledge. We denote the prior best response as the decision rule $\tilde{\pi}_0 := \text{BestResponseTo}(\mathcal{BP})$. The corresponding expected payoff pairs is denoted as $\boldsymbol{R}_0 = (R_0^i, R_0^j) = \psi(\cdot, \tilde{\pi}_0)$, since it is independent of $\tilde{\varphi}$. It means that the receiver has the ability to **unilaterally** determine both players' expected payoffs, which aligns with the intuition of the disagreement points in bargaining games.

The sender also possesses the unilateral ability to refuse cooperation by specifically choosing $\tilde{\varphi}$ such that $\mu = \mu_0$ (for instance, regardless of the current state $s$, the sender consistently sends a dummy signal $\sigma_0$, i.e., $\tilde{\varphi}(\sigma_0 \mid s) = 1, \forall s$). In this case, $\tilde{\pi}_1 = \tilde{\pi}_0$ and thus $\boldsymbol{R}_1 = \boldsymbol{R}_0$.

### 3.3 EXISTENCE ASSUMPTION OF BETTER OUTCOMES IN PERSUASION

The sender may not benefit from persuasion. That is, there is no guarantee that $R_1^i > R_0^i$ holds for any $\mathcal{BP}$.

**Proposition 3.1** (Paraphrased from Kamenica & Gentzkow (2011), Proposition 3). *If there does not exist a signaling scheme $\tilde{\varphi}$ such that, under the best response of a Bayesian receiver, the expected payoff of the sender is higher than the expected payoff under the receiver's best response to the prior distribution $\mu_0$, then the sender will not benefit from persuasion.*

The receiver may not benefit from persuasion either. In fact, (1) Based on the definition of $R_1^j$, the receiver utilizes Bayesian decision theory to guess a state $s'$ and optimizes its strategy accordingly. According to the properties of Bayesian decision theory, the error rate of guessing $s' = s$ using the posterior distribution $\mu$ is not greater than that of using the prior distribution $\mu_0$. (2) The upper bound of the receiver's expected payoff is when the sender honestly reports the state $s$, allowing the receiver to observe $s$ directly. Each time the receiver guesses incorrectly, its expected payoff does not increase; combining (1) and (2), it follows that $R_1^j \geq R_0^j$. We therefore need to argue under the following condition.

**Assumption 3.2** (Existence of Better Outcomes). *For a Bayesian persuasion $\mathcal{BP} = (\mathbb{S}, \mu_0, \mathbb{A}, r^i, r^j)$, there exists at least a strategy pair $(\tilde{\varphi}, \tilde{\pi})$ for the sender and receiver such that their respective expected payoffs exceed those at the disagreement point, i.e., $R^i(\tilde{\varphi}, \tilde{\pi}) > R^i(\tilde{\varphi}_0, \tilde{\pi}_0)$ and $R^j(\tilde{\varphi}, \tilde{\pi}) > R^j(\tilde{\varphi}_0, \tilde{\pi}_0)$.*

Similar arguments to our reduction will show that if Assumption 3.2 does not hold, then it is a bargaining game where they are to divide $0$ coin in Example 4.4.

## 3.4 ACTION RULE WITH GAME STRUCTURE AWARENESS

The receiver's awareness of the game structure is important, as it allows more complicated action rules. We define the **action rule with game structure awareness** as $\pi : \Psi \times \tilde{\Phi} \times \Sigma \to \Delta(\mathbb{A})$, which enables Receiver to decide its $\tilde{\pi}$ depending on both the game structure $\psi$ and the committed signaling scheme $\tilde{\varphi}$.

Consider an example $\pi$: Satisfaction Check, as described in Algorithm 2. The **satisfaction threshold function** is defined as an indicator function $\chi : \{\boldsymbol{R}\}^2 \to \{0, 1\}$ and reflects the receiver's non-myopic preferences, such as reflecting the receiver's demands for fairness. In the algorithm, the satisfaction threshold function is $\chi(\boldsymbol{R}_0, \boldsymbol{R}_1) = 1$ if and only if $R_1^i \leq R_1^j$, which means that the receiver will only respect the sender's recommendation if its posterior expected payoff is higher than the sender's. Then, if the receiver is satisfied with the committed signaling scheme, it will choose its best response action rule. Otherwise, the receiver will ignore all the sender's signals, and the sender will lose its influence and the potential benefits of its informational advantage.

Despite the introduction of the satisfaction threshold function may seem to suggest that the receiver is acting as a consequence of temperament or impulsive behavior, it aligns with the assumption of rationality. Detailed discussion is in Appendix C.

---

**Algorithm 2:** Satisfaction Check $\pi$: An Example Decision Rule with Game Structure Awareness

---

**Input:** A Bayesian persuasion task $\mathcal{BP} = (\mathbb{S}, \mu_0, \mathbb{A}, r^i, r^j)$, committed signaling scheme $\tilde{\varphi}$,
    received signal $\sigma$ ;
**Output:** An action distribution $b \in \Delta(\mathbb{A})$ ;
**Initialize:** Decide a satisfaction threshold function $\chi : \{\boldsymbol{R}\}^2 \to \{0, 1\}$ ;
**Initialize:** Reason out the game structure $\psi$ of $\mathcal{BP}$ ;
1 Calculate the best response to the posterior distribution $\tilde{\pi}_1 = \text{BestResponseTo}(\mu_0, \tilde{\varphi})$ ;
2 Ignore the signaling scheme and calculate the best response to the prior distribution
    $\tilde{\pi}_0 = \text{BestResponseTo}(\mu_0)$ ;
3 Calculate players' ground true expected payoffs $\boldsymbol{R}_0 = \psi(\tilde{\varphi}, \tilde{\pi}_0)$ and $\boldsymbol{R}_1 = \psi(\tilde{\varphi}, \tilde{\pi}_1)$ ;
4 **if** $R_1^i \leq R_1^j$ **then** $b = \tilde{\pi}_1(\cdot \mid \sigma)$ ;       // Satisfaction check. $\chi(\boldsymbol{R}_0, \boldsymbol{R}_1) = 1$.
5 **else** $b = \tilde{\pi}_0(\cdot \mid \sigma)$ ;
6 **return** $b$

---

The satisfaction threshold function is not unique. It could also depend on the committed signaling scheme $\tilde{\varphi}$, denoted as $\chi(\tilde{\varphi})$. An intuitive example is $\Sigma = \mathbb{S}$ and $\chi(\tilde{\varphi})$ returns $1$ if and only if the

sender is honest about the state, i.e., $\tilde{\varphi}(\sigma = s \mid s) = 1, \forall s$. We observe this satisfaction check rule implemented by LLM agents in the experiments (details in Appendix F).

### 3.5 Implications of Long-Term Interaction in Persuasion

The commitment assumption (Assumption 2.1) indicates that **Bayesian persuasion involves long-term interactions.** Without the implication of long-term interactions, the sender has no incentive to care about its reputation. It could tell the receiver that it would act upon one signaling scheme but in fact use another one. On the flip side, the receiver can also notice this possibility and will not trust the sender.

The underlying long-term nature of Bayesian persuasion enables the receiver to establish more sophisticated action rule $\pi$, by designing a satisfaction threshold. Justified by the sender's commitment assumption, it is also conceivable that the receiver has a commitment by disclosing its satisfaction function to the sender. If the receiver commits before the sender, then the sender must consider whether to fulfill the receiver's expectations. This is recognized as **the power of commitment** (Nowak et al., 2000). Detailed discussions can be found in Appendix C.

### 3.6 Joint Commitment

We define a joint commitment condition, where the sender and receiver will discuss to reach a consensus and then commit a strategy pair $(\tilde{\varphi}, \tilde{\pi})$. This strategy pair becomes common knowledge before the persuasion game starts, and then both players will play their committed strategy during the game.

**Definition 3.3** (Joint Commitment). *A joint commitment of a Bayesian persuasion task $\mathcal{BP} = (\mathbb{S}, \mu_0, \mathbb{A}, r^i, r^j)$ is a strategy pair $(\tilde{\varphi}, \tilde{\pi}) \in \tilde{\Phi} \times \tilde{\Pi}$, where the sender commits to $\tilde{\varphi}$ and the receiver commits to $\tilde{\pi}$, subject to the conditions that $\tilde{\varphi}^* \neq \tilde{\varphi}_0, \tilde{\pi}^* \neq \tilde{\pi}_0, \tilde{\Phi} := \{\tilde{\varphi} : \mathbb{S} \to \Delta(\Sigma)\}$, and $\tilde{\Pi} := \{\tilde{\pi} : \Sigma \to \Delta(\mathbb{A})\}$.*

**Definition 3.4** (Solution of Bayesian Persuasion). *A solution concept of Bayesian persuasion $\mathcal{BP} = (\mathbb{S}, \mu_0, \mathbb{A}, r^i, r^j)$ is a function $\omega_{\mathcal{BP}}$ which maps every $\mathcal{BP}$ to a joint commitment $(\tilde{\varphi}, \tilde{\pi}) \in \tilde{\Phi} \times \tilde{\Pi}$, where $\tilde{\Phi} := \{\tilde{\varphi} : \mathbb{S} \to \Delta(\Sigma)\}$ and $\tilde{\Pi} := \{\tilde{\pi} : \Sigma \to \Delta(\mathbb{A})\}$.*

### 3.7 Reduction from Bayesian Persuasion to Bargaining Games

**Lemma 3.5** (Reduction from Bayesian Persuasion to Bargaining Games). *When Assumption 3.2 holds, a Bayesian persuasion task $\mathcal{BP} = (\mathbb{S}, \mu_0, \mathbb{A}, r^i, r^j)$ is polynomial-time reducible to a bargaining game $\mathcal{BG} = (\mathbb{Y}, \boldsymbol{d})$. That is, $\mathcal{BP} \leq_p \mathcal{BG}$.*

*Proof.* With Assumption 3.2, there exists $\boldsymbol{R}$ such that $\boldsymbol{R} > \boldsymbol{R}_0$. Thus we can investigate the bargaining game where $\mathbb{Y} = \{\boldsymbol{R}\}$ and $\boldsymbol{d} = \boldsymbol{R}_0$. Given a bargaining solution concept $\omega_{\mathcal{BG}}$, one could obtain the solution of $\mathcal{BG}$, denoted as $\boldsymbol{y}_{\mathcal{BG}}$. Then we have $\boldsymbol{y}_{\mathcal{BG}} \in \mathbb{Y} = \{\boldsymbol{R}\}$. In this way, $\mathcal{BP}$ can be formulated as an optimization problem

$$\min_{(\tilde{\varphi}, \tilde{\pi})} 0 \quad \text{s.t.} \quad \boldsymbol{R} = \boldsymbol{y}_{\mathcal{BG}}. \tag{4}$$

By Equation (3), the optimization problem (4) is convex, which can be solved in polynomial time, as we desired. Denote the solved strategy pair as $(\tilde{\varphi}^*, \tilde{\pi}^*)$ and the induced expected payoffs as $\boldsymbol{R}^*$. Since $\boldsymbol{R} = \boldsymbol{y}_{\mathcal{BG}} \in \{\boldsymbol{R}\}$, we have $\boldsymbol{R}^* > \boldsymbol{R}_0$ by Definition 2.5. Thus $\tilde{\varphi}^* \neq \tilde{\varphi}_0$ and $\tilde{\pi}^* \neq \tilde{\pi}_0$, otherwise $\boldsymbol{R}^* = \boldsymbol{R}_0$. In this way, the strategy pair $(\tilde{\varphi}^*, \tilde{\pi}^*)$ is a joint commitment, which means we have found the solution of $\mathcal{BG}$. □

## 4 Empirical Results

In this section, we aim to demonstrate how the receiver in a BP task can use an action rule with game structure awareness to achieve higher expected gains than what the equilibrium of the canonical Bayesian persuasion formulation suggests. The experimental results reveal the receiver's retaliatory strategies.

## 4.1 TASKS OF PERSUASION AND BARGAINING

In this section, we describe several examples, where Bayesian persuasion equilibrium is a special case solution in bargaining games. These examples will later be the tested with LLM agents.

The grading student task, also known as recommendation letters, is introduced by Kamenica & Gentzkow (2011). We use the details provided in Dughmi (2017).

**Example 4.1** (Grading Students, GS, Dughmi (2017)). *Recent graduates entering the job market are divided into $1/3$ excellent ($s = 1$) students and $2/3$ weak ($s = 0$) students, and the distribution is $\mu_0$. A professor (the sender), who can assess student quality, sends grades as signals to an HR representative (the receiver) who makes hiring decisions. The professor gains $1$ reward per student hired, while the HR gains $1$ reward per excellent student hired and $-1$ reward per weak student hired. Neither party gains if a student remains not hired.*

The professor and the HR have no conflict when the current student is excellent, so the professor will report the state honestly in this case. Thus, the signaling scheme is $\tilde{\varphi}(\sigma = 1 \mid s = 1) = 1$ and $\tilde{\varphi}(\sigma = 1 \mid s = 0) = \eta$, where $\eta$ is a parameter $0 \leq \eta \leq 1$. Assuming the HR acts based on its posterior best response according to the canonical Bayesian persuasion, there are three example outcomes analyzed in Dughmi (2017): (1) The professor provides no information ($\eta = 1$). Their expected payoffs are both 0. (2) The professor is fully honest ($\eta = 0$). Their expected payoffs are both $1/3$. (3) The professor chooses an $\eta$ less than but very close to $1/2$. The expected payoffs of the professor and the HR are $(1 + 2\eta)/3$ and $(1 - 2\eta)/3$, respectively. The detailed analysis is in Appendix D.

The context of grading students has societal implications, because professors are expected to be more honest when providing information about their students. We therefore also test on an alternative example of Selling Products (SP), which mathematically has the same game structure, but experiences less social context.

**Example 4.2** (Selling Products, SP. A variant of Example 4.1.). *A seller (the sender) recommends its products to a buyer (the receiver). State is the quality of the product. The detailed value settings are the same with Example 4.1.*

The game structure of persuasion tasks, i.e. Example 4.1 and Example of 4.2, are equivalent to bargaining tasks, i.e. Asymmetric Bargaining (AB) in Example 4.3 and Splitting Coins (SC) in Example 4.4.

**Example 4.3** (Asymmetric Bargaining, AB). *A proposer suggests an $\eta \in [0, 1/2]$. If a responder agrees, the proposer receives a payoff of $(1+2\eta)/3$ and the responder receives a payoff of $(1-2\eta)/3$. If the responder disagrees, both get nothing.*

**Example 4.4** (Splitting Coins, SC). *Two players split $100$ coins. A Proposer suggests a split, and a Responder either accepts or rejects it. If accepted, they split as proposed; otherwise, both get nothing.*

## 4.2 EXPERIMENT SETUP WITH LARGE LANGUAGE MODELS

One way to test the behavior of intelligent agents in these tasks is to conduct human experiments. With the availability of LLMs, there are 3 main advantages of testing with LLMs instead of humans: (**1**) It is easy to provide long context to LLMs. In contrast, explaining the concept of persuasion and bargaining to an average human population may require significant effort. (**2**) It is immediate for LLMs to provide their interpretation as to why such decisions are made by giving appropriate prompts. (**3**) The experiments by LLMs scale much easier than human experiments, and are also more efficient in terms of monetary costs and environmental costs (e.g. carbon emission (Tomlinson et al., 2024)).

The tasks are tested with 3 variants regarding the farsightedness, namely, ($\alpha$) The agents play the game once, and were explicitly told that they will never meet again in the future. ($\beta$) The agents play the game once, but they are told that they may meet again in the future. ($\gamma$) The agents play the game multiple times. The maximum time $T$ is sampled by a memoryless distribution, which introduces **the shadow of the future** (Bó, 2005), preventing players from predicting when the game will end. We defer the exact prompt used to Appendix E.

| Task | Game | Setting | Sender's Payoff | Receiver's Payoff | Consensus Rate | $\eta$ |
|------|------|---------|-----------------|-------------------|----------------|--------|
| SC | $\mathcal{BG}$ | $\alpha$ | 93.30 (4.41) | 6.70 (4.41) | 1.00 | / |
| SC | $\mathcal{BG}$ | $\beta$ | 78.5 (15.47) | 21.5 (15.47) | 1.00 | / |
| SC | $\mathcal{BG}$ | $\gamma$ | 58.53 (3.80) | 41.47 (3.80) | 1.00 | / |
| AB | $\mathcal{BG}$ | $\alpha$ | 83.32 (37.9) | 0.01 (0.03) | 0.83 | 0.50 (0.00) |
| AB | $\mathcal{BG}$ | $\beta$ | 70.13 (25.09) | 19.87 (10.47) | 0.90 | 0.30 (0.10) |
| AB | $\mathcal{BG}$ | $\gamma$ | 74.46 (4.80) | 25.54 (4.80) | 1.00 | 0.24 (0.05) |
| GS | $\mathcal{BP}$ | $\beta$ | 0.00 (0.00) | 0.00 (0.00) | 0.00 | 0.38 (0.13) |
| SP | $\mathcal{BP}$ | $\beta$ | 0.00 (0.00) | 0.00 (0.00) | 0.00 | 0.45 (0.10) |
| GS | $\mathcal{BP}$ | $\gamma$ | 57.63 (4.12) | 42.37 (4.12) | 1.00 | 0.08 (0.04) |
| SP | $\mathcal{BP}$ | $\gamma$ | 58.00 (3.37) | 42.00 (3.37) | 1.00 | 0.08 (0.03) |

Table 1: Experiments solved by `gpt-4-turbo-2024-04-09`. The payoffs are the mean and standard deviation from 30 runs. Players' payoffs are scaled so that the maximum possible social welfare is 100, to facilitate comparison. The tasks SC, AB, GS, and SP are Splitting Coins, Asymmetric Bargaining, Grading Students, and Selling Products, respectively.

| Task | Game | Setting | Sender's Payoff | Receiver's Payoff | Consensus Rate | $\eta$ |
|------|------|---------|-----------------|-------------------|----------------|--------|
| SC | $\mathcal{BG}$ | $\alpha$ | 99.00 (0.00) | 1.00 (0.00) | 1.00 | / |
| SC | $\mathcal{BG}$ | $\beta$ | 79.37 (19.70) | 20.63 (19.70) | 1.00 | / |
| SC | $\mathcal{BG}$ | $\gamma$ | 51.60 (2.28) | 48.4 (2.28) | 1.00 | / |
| AB | $\mathcal{BG}$ | $\alpha$ | 65.93 (24.04) | 24.07 (12.07) | 0.90 | 0.26 (0.12) |
| AB | $\mathcal{BG}$ | $\beta$ | 71.00 (9.10) | 29.00 (9.10) | 1.00 | 0.21 (0.09) |
| AB | $\mathcal{BG}$ | $\gamma$ | 74.21 (2.51) | 25.79 (2.51) | 1.00 | 0.24 (0.03) |
| GS | $\mathcal{BP}$ | $\beta$ | 12.5 (28.43) | 4.17 (9.48) | 0.17 | 0.24 (0.00) |
| SP | $\mathcal{BP}$ | $\beta$ | 0.00 (0.00) | 0.00 (0.00) | 0.00 | 0.26 (0.05) |
| GS | $\mathcal{BP}$ | $\gamma$ | 61.72 (4.38) | 38.28 (4.38) | 1.00 | 0.12 (0.04) |
| SP | $\mathcal{BP}$ | $\gamma$ | 71.42 (5.40) | 28.58 (5.40) | 1.00 | 0.21 (0.05) |

Table 2: Experiments solved by `gpt-4o-2024-08-06`. The other settings are the same as Table 1.

Because the commitment assumption in persuasion tasks implicitly requires long-term interactions between the sender and receiver (details discussed in Section 3.5), setting ($\alpha$) will not apply to persuasion tasks. In the bargaining games of setting ($\gamma$), the initial proposer is uniformly chosen at random from among the two players. If the responder rejects the offer, the game continues with players switching roles; The game ends when the offer is accepted or the maximum time is reached. This setting is known as Rubinstein's alternating-offer model (Rubinstein, 1982). In persuasion games of setting ($\gamma$), the sender is always the proposer, and the receiver is always the responder. They will not switch the roles. The game ends when the receiver is satisfied with the committed signaling scheme or the maximum time is reached. This setting is similar to the model proposed by Gan et al. (2022).

All agents will have the common knowledge of the game settings. We conduct experiments on `gpt-4o-2024-05-13` and `gpt-4-turbo-2024-04-09` released by OpenAI (Achiam et al., 2023; Roumeliotis & Tselikas, 2023) through their API. Each configuration is tested with 30 independent runs. The total cost of querying the API of this work is $39.86 United States dollars.

### 4.3 EXPERIMENT RESULTS

Table 1 and 2 summarize the results of LLM agents playing the task games. We now discuss the results and the implications. More detailed output examples are given in Appendix F.

**Receiver's Retaliation and Anti-Exploitation.** In all bargaining and persuasion tasks, the proposer's offers given by LLM agents are incentive compatible, meaning that a myopic responder would accept it albeit the expected payoff they are offered. However, we notice that the receiver does not always accept: All cases where the consensus rate is not 1 are caused by the receiver's retaliatory action rule. The observed reasons are centered on that the receiver believes the sender's

signaling scheme results in too large a gap in their expected payoffs. E.g., in an outcome of SP-$\gamma$ by GPT-4-turbo, the buyer says that "Given the proposed signaling scheme where the seller scores bad products as good with a probability of 0.25, my expected payoff as Receiver is 0.167. This is significantly lower than the maximum possible payoff of 0.333 if no bad products are purchased. Therefore, I am not satisfied with this division as it heavily favors the seller." Interestingly, in the persuasion tasks GS-$\beta$ and SP-$\beta$ by GPT-4-turbo, as well as SP-$\beta$ by GPT-4o, the receiver rejected all signaling schemes, even if the proposed signaling scheme are essentially more fairer than the one in the canonical Bayesian persuasion analysis. It means under these settings the two instances of LLMs cannot effectively establish a deal between them.

**Effects of Long-Term Interaction.** For the proposer in most cases under setting ($\gamma$) by GPT-4o, it first proposes a much unfair offer and then slowly adjust to make it to be more satisfactory until the responder accepts it. For the responder in these cases, they reject the offers in the first few rounds, even in the tasks which demonstrate consensus rates of 1. In the end, they always agree on a compromised signaling scheme (the consensus rate reached its maximum value of 1). In GPT-4-turbo, the iterative setting, ($\gamma$), results in more fair outcomes than settings ($\alpha$) and ($\beta$), with the sender's signaling scheme being almost honest in the persuasion tasks GS-$\gamma$ and SP-$\gamma$.

We also notice that even if the game is played only once, the given context of the potential for future encounters can change their behavior. The proposer is more cautious in setting ($\beta$) by GPT-4o compared to setting ($\gamma$). E.g., in an outcome of SC-$\beta$ by GPT-4o, the buyer says that "I also need to consider that we might play this game again in the future, so I should offer a fair amount to ensure future cooperation. Offering the minimum amount that Responder would accept, such as 1 coin, might be risky if they expect a fairer split." In SC-$\beta$ compared to SC-$\alpha$, by GPT-4-turbo or GPT-4o, the sender's expected payoff decreased, while the receiver's payoff increased, and the consensus rate remained at its maximum value of 1. In AB-$\beta$ compared to AB-$\alpha$, by GPT-4-turbo or GPT-4o, the sender lowered their $\eta$, making the offer fairer, which resulted in an increase in the consensus rate and thus an improvement in social welfare (the sum of the proposer's payoff and the responder's payoff).

**Effects of the Task Context.** Different scenario contexts may induce different outcomes on the persuasion tasks. By both LLM solvers in settings ($\beta$) and ($\gamma$), the proposer tends to be fairer in Example 4.1 than in Example 4.2, even though these two examples are mathematically equivalent. In human social behavior, using the actions of people in a scenario to explain and predict beliefs is an important topic, where Bayesian theory of mind (Baker, 2012) studies how individuals update their beliefs based on the actions of others. The training process of LLMs includes reinforcement from human feedback, thereby aligning the models with human preferences. In tasks GS and SP, different contexts provided the LLM with varying prior beliefs about how the sender would perform. We speculate that the proposer may feel more pressure to lie when the situation involves human subjects (students). We tested the impact of these stereotypes on the identity of the sender in these scenarios, and the results showed that both GPT-4o and GPT-4-turbo perceived the professor as more honest than the salesperson: "The combination of ethical obligations, potential long-term consequences on reputation, and stronger normative frameworks in the academic setting of (GS) supports a higher likelihood of honest behavior compared to the commercial and competitive setting of (SP)." In reality, professors are expected to uphold academic integrity (Macfarlane et al., 2014) in their conduct. Sellers, on the other hand, do not have such explicit ethical constraints. Therefore, this phenomenon interestingly reflects how the contexts affect the persuasion processes.

## 5 CONCLUSION

We conclude that Bayesian persuasion is reducible to a bargaining game, which agrees with the diverse outcome of observation tasks in practice. This perspective offers a nuanced understanding of Bayesian persuasion, suggesting that signaling schemes may not only serve the sender's interests but also encourage the receiver to adopt retaliatory tactics. Our experimental evidence indicates that these strategies can lead to greater honesty from the sender, thus fostering more equitable interactions. Our results provide insights into the potential of the receiver to have anti-exploitation and counter-exploitation strategies, and could potentially be combined with other methods, such as institution, for improving equality and social welfare.

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

# A  GLOSSARY

$\Delta(\mathbb{X})$ denotes the simplex of the set $\mathbb{X}$. $\{x\}$ denotes the set whose elements are all the possible values of $x$.

| Symbol | Meaning | Note |
|---|---|---|
| $i$ | Sender's index | |
| $j$ | Receiver's index | |
| $s \in \mathbb{S}$ | Environmental state | |
| $\mu_0 \in \Delta(\mathbb{S})$ | Prior distribution of states | |
| $a \in \mathbb{A}$ | The receiver's action | |
| $\sigma \in \Sigma$ | The sender's signal | $\Sigma = \mathbb{A}$ with revelation principle (Proposition 2.2) |
| $\tau \in \mathbb{T}$ | The receiver's type | It can be seen by the sender but not by other receivers. |
| $q : \mathbb{S} \to \Delta(\mathbb{T})$ | Emission function | $q$ is w.l.o.g. to be dummy if there is only one receiver. |
| $\tilde{\Phi} := \{\tilde{\varphi} : \mathbb{S} \to \Delta(\Sigma)\}$ | The sender's signaling scheme set (without the game structure awareness) | The canonical setting in Bayesian persuasion |
| $\phi : \mathbb{S} \times \mathbb{T} \to \Sigma$ | The sender's signaling scheme if $\mathbb{T}$ is not dummy | |
| $\tilde{\Pi} := \{\tilde{\pi} : \Sigma \to \Delta(\mathbb{A})\}$ | The receiver's action rule set (without the game structure awareness) | The canonical setting in Bayesian persuasion |
| $\tilde{\pi}_0$ | The receiver's prior best response | The receiver ignores the sender |
| $\tilde{\pi}_1$ | The receiver's posterior best response | The default behavior in Bayesian persuasion |
| $r^i : \mathbb{S} \times \mathbb{A} \to \mathbb{R}$ | The sender's reward function | |
| $r^j : \mathbb{S} \times \mathbb{A} \to \mathbb{R}$ | The receiver's reward function | |
| $\mathcal{BP} := (\mathbb{S}, \mu_0, \mathbb{A}, r^i, r^j)$ | Bayesian persuasion task | |
| $\omega_{\mathcal{BP}}$ | A solution concept of Bayesian persuasion | It maps every $\mathcal{BP}$ to a strategy pair $(\tilde{\varphi}, \tilde{\pi}) \in \tilde{\Phi} \times \tilde{\Pi}$ |
| $\{R^i(\mu_0, \tilde{\varphi}, \tilde{\pi}, r^i)\} \subset \mathbb{R}$ | The sender's expected reward given $\mu_0, \tilde{\varphi}, \tilde{\pi}$, and $r^i$ | Simplified as $R^i(\tilde{\varphi}, \tilde{\pi})$ when $\mathcal{BP}$ is given and clear |
| $\{R^j(\mu_0, \tilde{\varphi}, \tilde{\pi}, r^j)\} \subset \mathbb{R}$ | The receiver's expected reward given $\mu_0, \tilde{\varphi}, \tilde{\pi}$, and $r^j$ | |
| $\boldsymbol{R} := (R^i(\tilde{\varphi}, \tilde{\pi}), R^j(\tilde{\varphi}, \tilde{\pi}))$ | Expected reward outcomes | $\boldsymbol{R}_0$ if $\tilde{\pi} = \tilde{\pi}_0$ and $\boldsymbol{R}_1$ if $\tilde{\pi} = \tilde{\pi}_1$ |
| $\Psi := \{\psi : \tilde{\Phi} \times \tilde{\Pi} \to \{\boldsymbol{R}\}\}$ | The game structure of a Bayesian persuasion task | Every $\psi$ is determined by a $\mathcal{BP}$ |
| $\chi : \Psi \times \tilde{\Phi} \to \{0, 1\}$ | The receiver's satisfaction threshold function | |
| $\Pi := \{\pi : \Psi \times \tilde{\Phi} \times \Sigma \to \Delta(\mathbb{A})\}$ | The receiver's action rule (with the game structure awareness) | E.g. $\pi_{\chi(\psi, \varphi)}$ |
| $\mathbb{V}^i$ | The sender's visibility set | It specifies what the sender can see |
| $\mathbb{V}^j$ | The receiver's visibility set | $\mathbb{V}^i \setminus \mathbb{V}^j = s$ |
| $\mathbb{V}^{i+j}$ | Players' common knowledge | Known by the players, with each aware that others know it, ad infinitum |

Table 3: Glossary of Bayesian Persuasion

| Symbol | Meaning | Note |
|--------|---------|------|
| $\mathbb{Y}$ | Feasibility set (or agreement set) | |
| $\boldsymbol{d}$ | Disagreement point | |
| $\mathcal{BG} := (\mathbb{Y}, \boldsymbol{d})$ | Bargaining game | |
| $\omega_{\mathcal{BG}}$ | A solution concept of bargaining games | It maps every $\mathcal{BG} = (\mathbb{Y}, \boldsymbol{d})$ to an agreement $\boldsymbol{y} \in \mathbb{Y}$ |

Table 4: Glossary of Bargaining Games

# B  RELATED WORK

## B.1  RELATED WORK ON BARGAINING

**Bargaining Games as Extensive-Form Games.**    Another perspective on solving the bargaining game is to formulate it as an extensive form game (Rubinstein, 1982). The proposer extends an offer, and the responder has the choice to either accept this offer or reject it. If accepted, the two parties reach a consensus and gain benefits; if rejected, both parties receive a default value (usually 0). If there is only one round, the responder will accept any offer greater than 0. In a multi-round scenario where the responder can propose offers after rejecting, this process repeats multiple times. After introducing a discount factor to represent patience levels, the expected outcomes for both parties will be a function of this discount factor. In particular, when both parties are infinitely patient (with a discount factor of 1), they will reach a deal in the first round and split the benefits equally. These solutions are referred to as subgame perfect equilibria.

**Human Experiments.**    Güth & Tietz (1990); Güth et al. (1982) found that considerations of distributive justice can significantly undermine strategic power in ultimatum bargaining. A more recent study by Lin et al. (2020) analyzed a large dataset, revealing that equal-split offers are accepted more frequently and quickly than slightly unequal ones, in both one-shot and repeated interactions.

**Evolutionary Game Theory.**    Another approach to understanding bargaining behavior is through evolutionary game theory. Nowak et al. (2000) developed an evolutionary model of the one-shot bargaining game, demonstrating that fairness can evolve if the proposer has access to information about past deals accepted by the responder. This suggests that the evolution of fairness is closely tied to reputation. Such insights highlight the dynamic nature of fairness in negotiations and its implications for strategic interactions.

**Language Models Experiments.**    An early attempt to explore bargaining using language models was made by He et al. (2018), who employed generative models to conduct bargaining experiments. They proposed a method to decouple the generation and decision-making processes. With the rise of large language models, recent studies have emerged focusing on their application in bargaining scenarios. Xia et al. (2024) introduced a benchmark for a buyer-seller bargaining game, demonstrating that OpenAI's ChatGPT performed well across various metrics among several large models. Another work (Deng et al., 2024) noted that "LLM agents can (naturally) be good negotiators," highlighting the potential of LLMs in negotiation contexts.

## B.2  EVIDENCE OF BAYESIAN PERSUASION BEING A BARGAINING GAME.

**The Hardness of Persuading a Far-sighted Receiver**    Gan et al. (2022) discusses the long-term interaction between the sender and receiver in an MDP, where both are aware of the realization of states in the MDP. The sender's informational advantage lies in knowing an external parameter that only affects their payoffs without impacting the MDP's transition function. In such a setting, persuading a far-sighted receiver is NP-hard. **Evidence (1)** They considered an easier setting where the goal is to persuade an advice-myopic receiver, who treats the sender as someone that will disappear in the future and makes decisions based solely on prior beliefs. They proved that persuading an advice-myopic receiver is solvable in polynomial time because the simplification of the receiver's strategy leads to certain properties in the MDP that can be simplified. Such a policy of the receiver is equivalent to a repeated bargaining game where the receiver commits to reject cooperation in the

future. **Evidence (2)** They introduced a threat-based meta signaling scheme for the sender, where if the sender finds that Receiver does not follow its recommendation, it will cease providing any information in the future. In this case, the receiver's best response is to follow the recommendation. This strategy of the sender is a grim-trigger policy (Friedman, 1971) in a repeated game, and threat-based strategies are also common in bargaining scenarios.

**Markov Signaling Games, MSGs.** (Lin et al., 2023) proposed another Markov process (MSG) in which only the sender can see the Markov state. The receiver has observations, but these observations are common knowledge among the players; thus, the sender's informational advantage is reflected in the difference between the Markov state and Receiver's observations. The main difference from Gan et al. (2022) is that the sender's informational advantage covers the MDP transitions, since the transition function depends on the Markov state. In the discussion of the method in (Lin et al., 2023), a threat strategy of the receiver is mentioned. To strengthen the sender's persuasion, they constrains the signaling scheme to a stronger obedience, ensuring that the recommended actions generate higher expected posterior payoffs for the receiver, thus satisfying it.

**History-Dependent Signaling Scheme.** Bernasconi et al. (2024) extends the setting of Gan et al. (2022) by letting the external parameter known exclusively to the sender determines the environment's state transitions. Their work demonstrates that Markovian signaling schemes are not optimal and introduces a convenient subset of history-dependent signaling schemes called promise-form. This approach encodes past interactions as honest promises about the receiver's future rewards, which can be understood as a strategy the sender uses to respond to bargaining situations.

**Markov persuasion processes (MPPs).** Another series of studies examines the Markov persuasion processes (MPPs) model (Wu et al., 2022). In this model, long-term interactions occur only between the sender and the Markovian environment. At each timestep, a new receiver interacts with the sender, leaves the system before the next timestep begins, and receives the corresponding reward. Each receiver is myopic and will only interact with the sender once. As a result, the players do not engage with the issues involved in repeated games, and there is no bargaining phenomenon.

### B.3 Other Sequential Persuasion Works

Alonso & Câmara (2018) investigated the trade-offs between the persuasiveness of expert advice and the reliability of the information provided. Celli et al. (2020) expanded on this by exploring the impact of private information on Bayesian persuasion in sequential game with imperfect information. Li & Norman (2021) extended classic Bayesian persuasion games to sequential Bayesian persuasion games with multiple senders. This sequential extension explored how information can be gradually revealed over time, enhancing strategic interaction in dynamic settings. Wu (2023) studied how the timing of information release and the order of persuasion efforts affect outcomes in sequential persuasion scenarios.

## C Commitment of the Receiver

### C.1 Satisfaction Commitment

The action rule $\pi : \Psi \times \tilde{\Phi} \times \Sigma \to \Delta(\mathbb{A})$ with game structure awareness allows us to define the commitment behavior for the receiver.

**Definition C.1** (Commitment of the Receiver)**.** *The receiver commits to satisfaction threshold function $\chi$, which encompasses the following three aspects: (1) The receiver will decide on a $\chi$ before the game starts, (2) The receiver will honestly inform the sender of this $\chi$, and (3) during the game, the receiver will act upon $\chi$ using Algorithm 2.*

Committing a satisfaction threshold function does not contradict the rationality assumption. A temporary refusal to cooperate might inform the sender that the receiver is dissatisfied with the expected payoff outcome of the current signaling scheme, thereby forcing the sender to reveal more information about the state in the future. Therefore, choosing to refuse cooperation could be beneficial for long-term expected payoff gains, making it a rational behavior.

## C.2 THE POWER OF COMMITMENT

In the canonical setting of Bayesian persuasion it assumes the commitment of the sender (Assumption 2.1) and the sender only. The setting is one-shot, which disagrees with implication of long-term interaction brought by the commitment assumption. The receiver then have to act upon Bayesian best response, for it being rational and knowing that $R_1^j \geq R_0^j$.

However, if we similarly assume the commitment of the receiver, e.g., with Algorithm 2, then the receiver turns to be the player who determines the set of possible outcomes. Intuitively, this is analogous to the receiver being the "proposer" in a bargaining process. The receiver will then only cooperate if the sender commits a $\tilde{\varphi}$ that induces an outcome that satisfies $\chi$. Knowing this, achieving incentive compatible requires providing higher payoff to the receiver for the sender. This effect agrees with **the power of commitment** described in bargaining games, as described in Nowak et al. (2000): "If the proposer has perfect knowledge of the responder's q value, then it is in fact the responder who makes the offer."

## D THE CANONICAL ANALYSIS OF EXAMPLE 4.1

If the current student is excellent ($s = 1$), the professor will report it honestly, since at this state they have no conflicts. We set $\tilde{\varphi}(1 \mid 1) = 1$ and $\tilde{\varphi}(0 \mid 1) = 0$.[3] Otherwise ($s = 0$), the professor tells the HR that the current student is excellent ($\sigma = 1$) with a probability of $\eta$, where $\eta \in [0, 1/2]$, i.e., $\tilde{\varphi}(1 \mid 0) = \eta$.

When the professor reports that the current student is weak ($\sigma = 0$), the HR would know that the student must be weak, for it can calculate its posterior belief $\mu(s = 0 \mid \sigma = 0) = 1$. So the HR would refuse to hire the student. Similarly, the HR calculates its posterior belief after receiving $\sigma = 1$ as $\mu(0 \mid 1) = 2\eta/(1 + 2\eta)$ and $\mu(1 \mid 1) = 1/(1 + 2\eta)$. Since $\eta \in [0, 1/2]$, we have $\mu(0 \mid 1) \leq \mu(1 \mid 1)$, so the HR will guess the current state is 1 when the sent signal is 1, according to the Bayesian decision rule. Following this, the best response of the HR is $\tilde{\pi}(0 \mid 0) = 1$ and $\tilde{\pi}(1 \mid 1) = 1$. And the expected payoffs of the professor and the HR are $(1 + 2\eta)/3$ and $(1 - 2\eta)/3$ respectively.

## E DETAILS OF EXPERIMENTS AND PROMPTS

### E.1 AGENT ASSUMPTION SYSTEM PROMPT

```
agent_assumption_system_prompt = """You are a self-interested rational
    player.

"Self-interested" means that you only care your own utilitarian payoffs,
    without necessarily considering the welfare of others. Even though
    sometimes you design your strategy depending on the other party's
    utility function, your ultimate goal is still to optimize your own
    expected payoffs.

"Rational" means that you will always choose the strategy that brings you
    a higher expected payoff. That is, given any two strategies A and B,
    if strategy A provides a higher expected payoff than strategy B, you
    will always choose strategy A over strategy B. Even if A brings only
    a small improvement.

Therefore, when making decisions, you need to compare and ensure that
    this strategy brings a higher expected payoff than any other strategy
    you could choose."""
```

### E.2 DETAILS OF SPLITTING COINS

#### E.2.1 SPLITTING COINS TASK DESCRIPTION

---

[3]The other possible setting is symmetric, i.e., $\tilde{\varphi}(1 \mid 1) = 0$ and $\tilde{\varphi}(0 \mid 1) = 1$.

```
1 bargaining_task_description_system_prompt = """And, apart from you, there
      is another player just like you, and you are going to play a
      bargaining game.
2
3 In this game, you have to divide 100 one-dollar coins. One of you will be
      the proposer and the other the responder. The proposer will suggest
      how to split the money, and the responder will decide whether to
      accept this proposal.
4
5 If the responder accepts the proposal, they will divide the money
      according to the proposal and each get its respective amounts. If the
       responder rejects the proposal, both players will get nothing.
6
7 Your objective is to collect as many coins as possible."""
```

### E.2.2   SETTINGS OF BARGAINING

```
1 ultimatum_system_prompt = """You two agents will only play this game once
      . You will not have any interaction with it afterwards. The offer
      proposed by the proposer is the final proposal. If the responder
      rejects, it means breaking the deal, and both parties end up with
      nothing. There is no communication between you."""
2
3 ultimatum_but_may_meet_again_system_prompt = """The offer proposed by the
       proposer is the final proposal. If the responder rejects, it means
      breaking the deal, and both parties end up with nothing. There is no
      communication between you. But note that you two agents might play
      this game again in the future."""
4
5 alternating_offer_system_prompt = """You two agents will play this game
      multiple times. In the beginning, whether you become the proposer or
      the responder is determined by uniform random selection. If the
      current proposal is rejected, the game continues with the roles
      switching (the current responder becomes the proposer and the current
       proposer becomes the responder) until a consensus is reached.
6
7 If a consensus is never reached, the game may stop at any time with a
      probability (the termination timestep is sampled from a memoryless
      distribution), and in this case, neither of you will know in advance
      when the game will end."""
```

### E.2.3   ROLE ASSIGNMENTS AND RECORDS IN BARGAINING

```
1 role_descriptions_system_prompt = """If you are the proposer: Please
      analyze how many coins you will keep for yourself, with the remainder
       going to the responder. Format the output in JSON according to the
      following template:
2 {
3     "Analysis": "(Your Summarized Analysis)",
4     "Proposer wants to keep": x
5 }
6 Where x should be an integer in the range [0,100].
7
8 If you are the responder: You will be given a proposal. Based on this,
      please think step by step to make a decision on whether to accept
      this proposal. Format the output in JSON according to the following
      template:
9 {
10    "Analysis": "(Your Summarized Analysis)",
11    "Responder's decision": "yes" or "no",
12 }
13
```

```
14 Please STRICTLY adhere to the JSON templates when outputting, and do not
      output anything else. DON'T FOGET THE CURLY BRACES.
15
16 Whether you are the proposer or the responder, please think step by step,
      and then make a decision. Remember that you and your opponents are
      both self-interested and rational players. Be aware of the
      consequences of your decisions. Your analysis and decisions should
      remain logically CONSISTENT."""
```

```
1 proposal_template = """Now, the proposer proposes to keep {proposer_coins
      } coins for itself, the responder gets the remaining {responder_coins
      } coins."""
2
3 proposer_role_assignment_user_prompt = """Now, you are the proposer.
      Please make a decision using the proposer's JSON output template
      based on all the information you know."""
4
5 responder_role_assignment_user_prompt = """Now, you are the responder.
      Please make a decision using the responder's JSON output template
      based on all the information you know."""
6
7 alternating_offer_history_record_prompt = """At timestep {timestep}: the
      proposer was agent {proposer_index} ({proposer_who}) and the
      responder was agent {responder_index} ({responder_who}).
8 At that time, agent {proposer_index} ({proposer_who}) proposed to keep {
      proposer_coins} coins, which would mean agent {responder_index} ({
      responder_who}) would get {responder_coins} coins.
9 And agent {responder_index} ({responder_who}) {decision_verb} the
      proposal."""
```

## E.3 DETAILS OF ASYMMETRIC BARGAINING

### E.3.1 ASYMMETRIC BARGAINING TASK DESCRIPTION

```
1 bargaining_task_description_system_prompt = """And, apart from you, there
      is another player just like you, and you are going to play a
      bargaining game.
2
3 In this game, one player acts as the proposer, suggesting a real number
      $x$ (where $0 \leq x \leq 0.5$), while the other player, the
      responder, decides whether to accept or reject the offer. If the
      responder accepts, the proposer receives a payoff of $(1+2*x)/3$, and
      the responder gets $(1-2*x)/3$. If the offer is rejected, both
      players receive nothing. Each player's objective is to maximize their
      own payoff."""
```

### E.3.2 OTHER ASYMMETRIC BARGAINING PROMPTS

```
1 # ========================================
2
3 role_descriptions_system_prompt = """If you are the proposer: Please
      analyze how should you specify the $x$.
4
5 Format the output in JSON according to the following template:
6 {
7     "Analysis": "(Your Summarized Analysis)",
8     "Proposer's offer": x
9 }
10 Where x should be a real number in the range [0, 0.5].
11
```

```
12  If you are the responder: You will be given a proposal. Based on this,
        please think step by step to make a decision on whether to accept
        this proposal. Format the output in JSON according to the following
        template:
13  {
14      "Analysis": "(Your Summarized Analysis)",
15      "Responder's decision": "yes" or "no",
16  }
17
18  Please STRICTLY adhere to the JSON templates when outputting, and do not
        output anything else. DON'T FOGET THE CURLY BRACES.
19
20  Whether you are the proposer or the responder, please think step by step,
         and then make a decision. Remember that you and your opponents are
        both self-interested and rational players. Be aware of the
        consequences of your decisions. Your analysis and decisions should
        remain logically CONSISTENT."""
21
22  # draft: If you do not accept, please state what your expectation is. If
        you accept, then your expectation should be the money in the
        currently accepted proposal.
23
24  # =========================================
25
26  proposal_template = """Now, the proposer proposes to specify $x$ as {x}.
        This means that the proposer will get a payoff of {proposer_payoff:.3
        f} ($(1+2*x)/3$), and the responder gets {responder_payoff:.3f} ($
        (1-2*x)/3$), if the responder accepts."""
27
28  proposer_role_assignment_user_prompt = """Now, you are the proposer.
        Please make a decision using the proposer's JSON output template
        based on all the information you know."""
29
30  responder_role_assignment_user_prompt = """Now, you are the responder.
        Please make a decision using the responder's JSON output template
        based on all the information you know."""
31
32  # =========================================
33
34  alternating_offer_history_record_prompt = """At timestep {timestep}: the
        proposer was agent {proposer_index} ({proposer_who}) and the
        responder was agent {responder_index} ({responder_who}).
35  At that time, agent {proposer_index} ({proposer_who}) proposed to specify
         $x$ as {x}, which would mean agent {proposer_index} ({proposer_who})
         would get {proposer_payoff:.3f} and agent {responder_index} ({
        responder_who}) would get {responder_payoff:.3f}.
36  And agent {responder_index} ({responder_who}) {decision_verb} the
        proposal."""
```

### E.4 DETAILS OF GRADING STUDENTS

#### E.4.1 GRADING STUDENTS TASK DESCRIPTION

```
1  persuasion_student_task_description_system_prompt = """And, apart from
       you, there is another player just like you, and you are going to play
        a persuasion game.
2
3  In this agme, some recent graduates are entering the job market. Of these
        graduates, one third (1/3) are excellent, and two thirds (2/3) are
       weak.
4
5  A professor can directly see the students' qualities; And an HR can
       decide whether to hire based on the grades given by the professor. In
```

```
      this sense, the professor acts as the sender and the HR as the
      receiver, with the grade of each student serving as the sender's
      signal.

Both the professor and the HR representative are aware of this quality
      distribution, which is common knowledge between them. That is, they
      are both aware of the distribution and know that the other is aware
      as well, continuing ad infinitum.

The professor's goal is to maximize the number of students hired, as each
       hire yields a reward. Conversely, the HR aims to hire as many
      excellent students as possible, gaining a reward for each excellent
      student hired and incurring a penalty for each weak student hired.
      There is no reward or penalty for both players if a student is not
      hired.

Both parties strive to maximize their rewards."""
```

### E.4.2 COMMITMENT ASSUMPTION

```
commitment_student_system_prompt = """We assume the professor's grading (
      Sener's signal space) is binary; a score of 0 indicates the student
      is weak, and a score of 1 indicates the student is excellent. The
      sender's signaling scheme specifies the probabilities of assigning a
      score of 1 to both good and bad students.

The professor (Sender) will commit to a signaling scheme before the games
       begin and will honestly tell it to the HR (Receiver). Throughout the
       game, the professor will strictly adhere to the agreed-upon
      signaling scheme.

The emergence of this commitment is driven by the professor's need to
      build a credible reputation to optimize long-term payoff expectations
      . The sender and receiver will interact multiple times, such as due
      to multiple students and future graduations."""
```

### E.4.3 SETTINGS OF GRADING STUDENTS

```
ultimatum_but_may_meet_again_system_prompt = """The task for the sender
      is to propose a signaling scheme, and then the receiver should decide
       its action rule based on it. You two players cannot communicate. The
       signaling scheme proposed by the sender is the final decision, and
      the HR can only respond based on it. But note that YOU TWO AGENTS
      WILL PLAY THIS GAME AGIAN IN THE FUTURE."""

alternating_offer_system_prompt = """You two agents will play this game
      multiple times. In the beginning, the sender should propose a
      signaling scheme, and then the receiver should decide its action rule
       based on it.

And then the receiver will report whether it is satisfied with this
      outcome. If it is not satisfied, then the sender should propose a new
       signaling scheme and the game goes like previously described. If it
      is satisfied, then the game is over and the players get the
      corresponding rewards.

If a consensus is never reached, the game may stop at any time with a
      probability (the termination timestep is sampled from a memoryless
      distribution), and in this case, neither of you will know in advance
      when the game will end."""
```

### E.4.4 REVELATION PRINCIPLE

Sender and Receiver are to reach a consensus on the joint commitment. Sender will first commit a signaling scheme, and then Receiver will report whether it is satisfied with this, with along deciding its action rule. Players are given the context and the game structure $\psi(\eta) = \boldsymbol{R} = ((1+2\eta)/3, (1-2\eta)/3)$.

```
revelation_student_system_prompt = """When the current student is
    excellent (state=1), then the sender has no conflict with Receier, so
     the sender will report the state honestly, i.e., the sender will
    score 1 with probability of 1 when state=1.

However, there is a slight conflict between the sender and the receiver
    when the current student is weak (state=0): the sender wants the
    receiver to hire more students no matter of the student's quality,
    while the receiver does not want to hire weak students. So the sender
     may lie about the weak student's quality, scoring weak students as 1
     with a probability of $\eta$, expecting the receiver to hire it.

In this sense, the sender's signaling scheme is fully parameterized by $\
    eta$, and $0 \leq \eta \leq 0.5$. And the expected payoffs of the
    sender and the receiver are $(1+2*\eta)/3$ and $1-2*\eta$,
    respectively."""
```

### E.4.5 ROLE ASSIGNMENTS AND RECORDS IN PERSUASION

```
sender_proposal_user_prompt = """Now you are the sender, and you need to
    decide the value of $\eta$ to specify your signaling scheme. You
    should decide by considering the game structure, which is a mapping
    from sender's $\eta$ to sender's payoff and receiver's payoff, which
    are $(1+2*\eta)/3$ and $(1-2*\eta)/3$, repectively.

Format the output in JSON according to the following template:
{
    "Analysis": "(Your Summarized Analysis)",
    "Signaling Scheme": \eta
}
Where $0 \leq \eta \leq 0.5$.

Please STRICTLY adhere to the JSON templates when outputting, and do not
    output anything else. Do not use Markdown code block syntax; just
    output the JSON content directly."""

proposal_template = """Now the sender proposed to score 1 with
    probability of 1 when state=1 and score 1 with probability of $\eta={
    eta}$ when state=0. This means if the receiver acts upon its Bayesian
     best response, the sender will get a payoff of {sender_payoff:.3f}
    and the receiver will get a paoff of {receiver_payoff:.3f}. The game
    structure is a mapping from sender's $\eta$ to sender's payoff and
    receiver's payoff, which are $(1+2*\eta)/3$ and $(1-2*\eta)/3$,
    repectively."""

receiver_check_student_user_prompt = """Now you are the receiver. Given
    the proposal, please state whether you are satisfied with this
    division, and then specify your action rule.

You don't need to do any additional calculation such as calculating a
    posterior belief, for it is included in the result of the game
    structure; please make a decision based on all the currently known
    information.

Your action rule should remain CONSISTENT with your judgment of whether
    you are satisfied. That is, if you are satisfied, then you follow
    Sender's recommendation, otherwise not.
```

```
19
20 Format the output in JSON according to the following template:
21 {
22     "Analysis": "(Your Summarized Analysis)",
23     "Satisfied with the current committed signaling scheme and its
            corresponding reward outcome": "yes" or "no",
24     "Probability of hiring upon scoring 0": x,
25     "Probability of hiring upon scoring 1": y
26 }
27 Where x and y should be real numbers in the range [0,1].
28
29 Please STRICTLY adhere to the JSON templates when outputting, and do not
       output anything else. Do not use Markdown code block syntax; just
       output the JSON content directly."""
30
31 # =========================================
32
33 alternating_offer_history_record_prompt = """"At timestep {timestep}: the
       sender proposed to score 1 with probability of 1 when state=1 and
       score 1 with probability of {eta} when state=0. This means the sender
        wanted to keep a payoff of {sender_payoff:.3f} and left the receiver
        with a paoff of {receiver_payoff:.3f}. The receiver {receiver_verb}
       the proposal."""
```

### E.5 DETAILS OF SELLING PRODUCTS

### E.5.1 SELLING PRODUCTS TASK DESCRIPTION

```
1 persuasion_product_task_description_system_prompt = """And, apart from
      you, there is another player just like you, and you are going to play
       a persuasion game.
2
3 In this market, a variety of products are available for sale. Of these
      products, one third (1/3) are of good quality, and two thirds (2/3)
      are of bad quality.
4
5 A seller can directly see the quality of their products; And a buyer
      decides whether to purchase based on the signals provided by the
      seller. In this sense, the seller acts as the sender and the buyer as
       the receiver.
6
7 Both the seller and the buyer are aware of this quality distribution,
      which is common knowledge between them. That is, they are both aware
      of the distribution and know that the other is aware as well,
      continuing ad infinitum.
8
9 The seller's goal is to maximize the number of products sold, as each
      sale yields a reward. Conversely, the buyer aims to purchase as many
      good products as possible, gaining a reward for each good product
      purchased and incurring a penalty for each bad product purchased.
      There is no reward or penalty for both players if a product is not
      purchased.
10
11 Both parties strive to maximize their rewards."""
```

## F SEVERAL DETAILED EXAMPLE RESULTS

A run of the experiments SP-$\gamma$ by GPT-4-turbo:

```
=====================================================
i:  4
Elapsed Time of query_GPT: 4.741272211074829s
```

```
Agent 1:

{
    "Analysis": "As the proposer, I aim to maximize my share of the coins
        while ensuring the proposal is accepted by the responder. Given
        that both players are rational and self-interested, the responder
        will accept any proposal where they receive a non-zero amount,
        as rejecting it risks ending up with nothing, especially at the
        initial stages of the game where the risk of the game terminating
        is lower. To optimize my payoff while maintaining a high
        probability of acceptance, I propose keeping a majority of the
        coins for myself while offering just enough to make acceptance
        appealing to the responder.",
    "Proposer wants to keep": 99
}
Proposal:
 Now, the proposer proposes to keep 99 coins for itself, the responder
     gets the remaining 1 coins.
.................................................
Elapsed Time of query_GPT: 5.2775750160217285s
Agent 0:

{
    "Analysis": "As the responder, I must decide whether to accept or
        reject a proposal where I receive 1 coin and the proposer
        receives 99 coins. Given that the game could end at any time with
         both players receiving nothing, and considering that rejecting
        the offer switches roles, allowing me to propose a potentially
        more favorable split, I must weigh the risk of the game ending
        with zero payoff against the potential gain in a future round.
        However, since this is the first timestep and the game is
        unlikely to end immediately, I have an incentive to reject the
        proposal, aiming for a more balanced offer in the subsequent
        round where I will be the proposer.",
    "Responder's decision": "no"
}
-------------------------------------------------
Elapsed Time of query_GPT: 4.99947190284729s
Agent 0:

{
    "Analysis": "As the proposer, I aim to maximize my share while
        ensuring the proposal is accepted by the responder, who is also
        rational and self-interested. Given the previous rejection of a
        99-1 split by the responder, it indicates a preference for a more
         equitable distribution. To increase the likelihood of acceptance
         and maximize my gain, I propose a split that offers the
        responder slightly more than the minimum they might accept,
        considering the risk of game termination and role reversal which
        could lead to a less favorable proposal for me in future rounds
        .",
    "Proposer wants to keep": 60
}
Proposal:
 Now, the proposer proposes to keep 60 coins for itself, the responder
     gets the remaining 40 coins.
.................................................
Elapsed Time of query_GPT: 4.446723222732544s
Agent 1:

{
    "Analysis": "As the responder, I need to decide whether to accept the
         proposal of receiving 40 coins while the proposer keeps 60.
        Given that the game could end at any time with both players
        receiving nothing, and considering the previous proposal where I
```

```
                was offered only 1 coin, 40 coins is a significantly better offer
                . Accepting this proposal guarantees a substantial amount,
                whereas rejecting it risks ending up with nothing if the game
                terminates unexpectedly. Since my goal is to maximize my payoff,
                accepting this offer is rational.",
        "Responder's decision": "yes"
}
--------------------------------------------------
```

A run of the experiments SP-$\gamma$ by GPT-4-turbo:

```
==================================================
i:  18
Elapsed Time of query_GPT: 6.673327922821045s
{
    "Analysis": "To maximize my payoff as the sender, I need to choose an
        $\eta$ that balances the increase in my expected payoff with the
        risk of the receiver rejecting the scheme due to their
        decreasing payoff. The sender's payoff is maximized when $\eta$
        is maximized, but a higher $\eta$ reduces the receiver's payoff,
        increasing the likelihood of rejection. Since the receiver's
        payoff becomes negative when $\eta > 0.25$, setting $\eta$ close
        to this threshold maximizes my payoff while keeping the receiver'
        s payoff non-negative, thus maintaining a balance that might be
        acceptable to the receiver.",
    "Signaling Scheme": 0.25
}
...................................................
Elapsed Time of query_GPT: 5.121338129043579s
{
    "Analysis": "Given the proposed signaling scheme where the seller
        scores bad products as good with a probability of 0.25, my
        expected payoff as the receiver is 0.167. This is significantly
        lower than the seller's payoff of 0.500. Since my goal is to
        maximize my own payoff, I am not satisfied with this division as
        it disproportionately favors the seller.",
    "Satisfied with the current committed signaling scheme and its
        corresponding reward outcome": "no",
    "Probability of buying upon scoring 0": 0,
    "Probability of buying upon scoring 1": 0
}
--------------------------------------------------
Elapsed Time of query_GPT: 8.328505039215088s
{
    "Analysis": "To maximize my payoff as the sender, I need to consider
        the balance between selling as many products as possible and
        maintaining a credible reputation to ensure the receiver is
        willing to buy. The receiver's satisfaction is crucial for the
        game to end favorably for me. Given the receiver rejected an $\
        eta$ of 0.25, indicating a preference for higher quality
        assurance, I should propose a lower $\eta$ to increase the
        likelihood of the receiver accepting the signaling scheme. A
        lower $\eta$ decreases my potential payoff slightly but increases
         the receiver's payoff, making acceptance more likely. I will
        choose $\eta = 0.1$ to offer a better balance, providing me a
        payoff of $(1+2*0.1)/3 = 0.4$ and the receiver a payoff of $
        (1-2*0.1)/3 = 0.2667$.",
    "Signaling Scheme": 0.1
}
...................................................
Elapsed Time of query_GPT: 6.5160088539123535s
{
    "Analysis": "Given the proposed signaling scheme where the sender
        scores bad products as good with a probability of 0.1, the
        expected payoff for the receiver is 0.267. This is calculated
```

```
              based on the formula (1-2*eta)/3. Although this is not the
              maximum possible payoff for the receiver (which would be 0.333
              when eta=0), it still provides a positive expected value for good
               products outweighing the penalty from bad products. The sender's
               honesty in scoring good products as good (probability 1) and
              relatively low dishonesty rate for bad products makes this scheme
               somewhat acceptable.",
       "Satisfied with the current committed signaling scheme and its
          corresponding reward outcome": "yes",
       "Probability of buying upon scoring 0": 0,
       "Probability of buying upon scoring 1": 1
}
----------------------------------------------------
Final decisions:
  {
     "signaling scheme": {
         "prob_score1_bad": 0.1,
         "prob_score1_good": 1
     },
     "action rule": {
         "prob_buy_score0": 0,
         "prob_buy_score1": 1
     }
}
Sender's expected payoff: 0.400
Receiver's expected payoff: 0.267
Social welfare: 0.667
```

A run of the experiments SC-$\beta$ by GPT-4o:

```
{
    "Analysis": "As the proposer, I need to make an offer that the
        responder will accept, while maximizing my own payoff. The
        responder is rational and self-interested, so they will accept
        any offer that gives them more than they would get if they reject
        , which is zero. However, I also need to consider that we might
        play this game again in the future, so I should offer a fair
        amount to ensure future cooperation. Offering the minimum amount
        that the responder would accept, such as 1 coin, might be risky
        if they expect a fairer split. A common strategy is to offer a
        50-50 split, but I can also try offering slightly less to see if
        they accept, such as 40 coins for them and 60 for me.",
    "Proposer wants to keep": 60
}
```

A run of the experiments SC-$\beta$ by GPT-4o:

```
{
    "Analysis": "As the proposer, I need to make an offer that the
        responder will accept, while maximizing my own payoff. The
        responder, being rational and self-interested, will accept any
        offer that gives them more than they would get if they reject,
        which is zero. However, I must also consider that the responder
        might reject offers they perceive as unfair, especially if we
        might play this game again in the future. A common strategy is to
         offer a split that is slightly in favor of the proposer but
        still reasonable for the responder. Offering 60 to myself and 40
        to the responder is a common equilibrium in such games, as it
        provides the responder with a significant amount while maximizing
         my own payoff.",
    "Proposer wants to keep": 60
}
```

A run of the experiments AB-$\beta$ by GPT-4o:

```
{
    "Analysis": "As the proposer, I need to choose an x that maximizes my
        payoff while ensuring the responder accepts the offer. The
        responder will accept if their payoff (1-2*x)/3 is greater than
        0, which means x must be less than or equal to 0.5. However,
        since the responder is rational and self-interested, they will
        accept the smallest x that gives them a positive payoff.
        Therefore, I should choose the smallest x that gives me a
        positive payoff and is likely to be accepted by the responder.
        The smallest positive x that ensures the responder gets a
        positive payoff is slightly above 0, but to ensure acceptance and
         maximize my payoff, I should choose a slightly higher x that
        still gives me a good payoff. A reasonable choice is x = 0.25,
        which gives me a payoff of (1+2*0.25)/3 = 0.5 and the responder a
         payoff of (1-2*0.25)/3 = 0.25.",
    "Proposer's offer": 0.25
}
```

A run of the experiments AB-$\beta$ by GPT-4-turbo:

```
{
    "Analysis": "As a rational and self-interested responder, I must
        compare the payoff of accepting the proposal to the payoff of
        rejecting it. Accepting the proposal gives me a payoff of 0.007,
        while rejecting it results in a payoff of 0. Since 0.007 is
        greater than 0, accepting the proposal maximizes my immediate
        payoff. However, considering that we will play this game again in
         the future, accepting such an unfavorable offer might set a
        precedent for similarly low offers in subsequent games. This
        could potentially lead to lower overall future payoffs. Therefore
        , I must weigh the immediate benefit against the potential long-
        term disadvantage. Given the extreme disparity in the proposed
        distribution, rejecting the offer might encourage a more
        equitable proposal in future rounds, potentially increasing my
        long-term payoff.",
    "Responder's decision": "no"
}
```

A run of the experiments SP-$\gamma$ by GPT-4-turbo:

```
{
    "Analysis": "Given the proposed signaling scheme where the seller
        scores bad products as good with a probability of 0.25, my
        expected payoff as the receiver is 0.167. This is significantly
        lower than the maximum possible payoff of 0.333 if no bad
        products are purchased. Therefore, I am not satisfied with this
        division as it heavily favors the seller.",
    "Satisfied with the current committed signaling scheme and its
        corresponding reward outcome": "no",
    "Probability of buying upon scoring 0": 0,
    "Probability of buying upon scoring 1": 0
}
```

