# OpenReview forum: "Bayesian Persuasion Is a Bargaining Game"
_ICLR.cc/2025/Conference — Submitted to ICLR 2025_

### Official Review · Reviewer_RVe3 · 2024-11-02

**Soundness:** 1
**Presentation:** 2
**Contribution:** 2
**Rating:** 3
**Confidence:** 4

**Summary:**

This paper considers the problem of Bayesian persuasion (also known as "information design") in which an informed sender must transmit a signal to an uniformed receiver in order for the receiver to take an action that the sender favours. The main idea of the paper is that in some settings it can be reasonable to view not only the sender but also the receiver of having the ability to commit to their strategy in advance. In such cases, Bayesian persuasion can be viewed as a bargaining game. The authors discuss this idea and conduct some experiments with LLM agents to investigate the relevance of repeated interactions to both Bayesian persuasion and bargaining games.

**Strengths:**

As far as I am aware (though I am unfortunately not an expert on this literature), this paper is the first to discuss the possibility of receivers being able to make commitments in Bayesian persuasion settings and the implications of this. I think this is natural idea and one that -- as the authors argue -- does map on to several real-world scenarios. A better theoretical and empirical understanding of this setting could prove useful in reasoning about such scenarios. The definitions and explanations of both Bayesian persuasion and bargaining problems are reasonably clear (though see below for weaknesses), as is the explanation of the empirical results. The use of LLMs in their experiments is valid and -- as LLMs and related models mediate more and more of humans' interactions -- an increasingly interesting topic of study, in my opinion.

**Weaknesses:**

The primary issue I have with the paper is that in my view it purports to be doing something it is not. In several places (see, e.g., the first sentence of the conclusion) the authors claim that "Bayesian persuasion is reducible to a bargaining game", and indeed Lemma 3.5 is a formal version of this statement. I was quite confused for most of the paper however, because what the authors _actually_ show is that **a variant of Bayesian persuasion where the receiver is also given the ability to commit and the agents are able to interact beforehand** is reducible to a bargaining game. Put this way, the surprisingness and significance of the main claim (and in fact the _title_) of the paper is greatly diminished, in my opinion. This explains as well, for example, why the "polynomial time reduction" in Lemma 3.5 is so overblown. Specifically, the authors reduce Bayesian persuasion to a bargaining game via the formation of a constrained convex optimisation problem where nothing is being optimised! Instead, the reduction results from setting the solution of the first problem to the solution of the second in the constraint of the problem. At best this is naive, at worst it is deliberately misleading (in aiming to complicate and obfuscate the result so that it appears more significant than it actually is).

The other issues I have with the paper are less major, but are consistent with the problems above.

**Game-Theoretic Reasoning:** There are a few places where the game-theoretic reasoning in the paper seems confused:

- The authors claim that the ability of the _sender_ to commit implies that Bayesian persuasion involves long-term interactions, but this is not necessarily true -- there are plenty of one-time interactions where commitment is possible (e.g., contracts, escrow services, etc.). This is more of an issue as it this idea that the authors seem to motivate most of their experiments with. I suggest the authors weaken their claim accordingly.
- The authors claim in several places that it is game-theoretically rational for the receiver to refuse to best respond to certain commitments and signals provided by the sender, but this results only as a function of their assumption that the sender has a way of credibly committing themselves. Otherwise, the receiver is clearly making a non-credible threat and playing irrationally.

**Experimental Motivation:** There are a few things, in my mind, which make the experiments section weaker than it needs to be:

- The authors should clearly state up front what hypotheses they are aiming to test with their experiments and why. This is also related to the use of LLMs. The justification the authors provide for this choice seems to suggest that they are primarily interested in studying human behaviour and that they use LLMs as a proxy for this, but if so they should make this idea clearer. I happen to think that it is interesting in its own right to see how LLMs tackle such problems, especially if those problems can be linked to current or future real-world uses of LLMs, but the authors do not attempt to provide such links.
- Because of their mistaken supposition that the sender's ability to commit _requires_ long-term interactions, they do not consider the one-shot version (setting $\alpha$, in the paper's notation) Bayesian persuasion in their experiments, which is the obvious baseline that I was expecting to see.
- Why is the proposer chosen uniformly at random in the bargaining problems but not the sender in the Bayesian persuasion settings? More generally, at the moment these settings are examined side by side, but what should be the takeaways from the similarities and differences from the results in the two settings? What specific hypothesis has been confirmed or refuted (or neither) based on the similarities and differences between the results in the two settings?

**Notation:** This is overly complex and what appear to be typos early on in the paper make the Bayesian persuasion setting harder to understand than is necessary. Concretely:

- If the receiver has no private information, there is no need to talk about this (i.e., $q$ can be dropped completely). This also means that there is no need to distinguish between $\phi$ and $\varphi$, which would also save confusion.
- In footnote 1, it seems as though $q$ and $\phi$ are the wrong way round (the sender determines $\phi$, not $q$).
- The set of signalling schemes is denoted as blackboard bold $\psi$ but the signalling schemes themselves are elsewhere denoted as either $\phi$ or $\varphi$. Later $\psi$ is what determines the payoffs of the agents as a function of their commitments.
- Why do some symbols have tildes and others not? This is never explained.
- In Definition 3.3 $\tilde{\varphi}^*$ and $\tilde{\pi}^*$ have not been introduced.

**Language/Typos:** This is a lot more minor and I understand that the authors may not have English as a first language, but there are few claims that I could not understand because of the unusual choice of words:

- Page 1: What are "cognitive misers"?
- Page 3: What does it mean for "solutions to become conceivable"?
- Page 6: The sentence after Assumption 3.2 was unclear to me (in part because the example with the coins had not yet been introduced).
- Page 8: I think I understand what the authors mean here, but I would change the phrasing about different game structures "experienc[ing] less social context" to something like different settings being "less influenced by social norms and contexts not captured by the structure and payoffs included in the game description".

There are many other typos throughout and minor grammatical errors throughout which do not majorly impact the scientific quality and clarity of the paper, and so do not impact my review score, though I would nonetheless encourage the authors to spend a little more time to polish the manuscript in this regard.

**Questions:**

Please see the Weaknesses section for my questions. I also welcome the authors to correct any misunderstandings I may have about the paper and its claims.

---

> ### Author Response · Authors · 2024-11-18
> **Official Comment (RVe3 - 1) by Authors**
>
> Thank you for your detailed review and the many valuable suggestions for improvement. Regarding your primary issue, we acknowledge its significance, which aligns with reviewer giTx's concern. As such, we have addressed it in the public response; please kindly refer to that. Here, we will address the detailed issues point by point.
>
> **Game-Theoretic Reasoning (1): Long-Term Interactions**
>
> > **Reviewer RVe3:** The authors claim that the ability of the _sender_ to commit implies that Bayesian persuasion involves long-term interactions, but this is not necessarily true.
>
> We followed this explanation: "This is because in such settings, the principal maximizes his long-term utility by establishing a reputation for credibility." [1]  However, after double-checking, we found alternative justifications [2], such as the transparency of grading mechanisms in schools and the public enforcement of traffic laws. We agree with the reviewer that the justification for the commitment assumption extends beyond long-term interactions. **We will weaken our claims and include experiments under the $\alpha$ setting.**
>
> 1. Dughmi, Shaddin. "Algorithmic information structure design: a survey." *ACM SIGecom Exchanges* 15.2 (2017): 2-24.
> 2. Kamenica, Emir. "Bayesian persuasion and information design." *Annual Review of Economics* 11.1 (2019): 249-272.
>
> **Game-Theoretic Reasoning (2): Non-Credible Threat**
>
> > **Reviewer RVe3:** The authors claim in several places that it is game-theoretically rational for the receiver to refuse to best respond to certain commitments and signals provided by the sender, but this results only as a function of their assumption that the sender has a way of credibly committing themselves.
>
> This is an excellent point, and we suggest it highlights why our claim has not been addressed in the current literature. This relates to the development of bargaining games, as detailed in Appendix B.1. **We regard classical BP as an ultimatum game, where the analysis corresponds to subgame perfect equilibrium.**
> - From this perspective, the bargaining responder will accept any offer that gives them a payoff greater than their payoff from the disagreement point. For example, in the case of dividing 100 coins, this means that a proposal where the proposer takes 99 coins and the responder takes 1 coin would be accepted.
> - Correspondingly, the persuasion receiver will best respond to any signaling scheme that satisfies the obedience constraints. Therefore, the canonical analysis of the standard BP is reasonable.
>
> However, in human bargaining experiments, such extreme outcomes are rare, even when participants are informed they will only play once. **Another strand** of bargaining games considers **axiomatic** solution concepts, and our definition of BP and its reduction rely on this perspective. The axiomatic bargaining solution is a method that **uses mathematical rules (axioms) to define and solve allocation problems, focusing on whether the outcome is fair and reasonable,** rather than analyzing the bargaining process. It determines a unique solution by satisfying specific rules, such as **Pareto efficiency and symmetry,** exemplified by the well-known Nash bargaining solution.
>
> It is worth noting that even with a clear game dynamic, such as Rubinstein's alternating-offer model, agents tend to reach an equal split in the case of long-term interactions.[1] This is why we are so concerned with the literature on long-term interaction.
>
> What we focus on is the insights gained from viewing BP from the perspective of a bargaining game. Much of the BP community currently focuses on long-term interaction scenarios. Some recent work, which we discuss in B.2, can be interpreted within the framework of bargaining games. For example, certain studies strengthen the sender’s bargaining power by enabling them to refuse future information sharing if the receiver disregards their advice.
>
> **Reference**
>
> 1. Rubinstein, Ariel. "Perfect equilibrium in a bargaining model." _Econometrica: Journal of the Econometric Society_ (1982): 97-109.

---

> > ### Author Response · Authors · 2024-11-18
> > **Official Comment (RVe3 - 2) by Authors**
> >
> > **Experimental Motivation**
> >
> > Regarding the motivation for the experiment, an unified response has been provided in Public Response (2). Here, we address the specific questions.
> >
> > > **Reviewer RVe3:** The authors should clearly state up front what hypotheses they are aiming to test with their experiments and why. This is also related to the use of LLMs.
> >
> > Our goal in employing large models for BP is to demonstrate that receivers exhibit retaliatory behavior in BP, similar to behavior observed in BG, especially under implicit or explicit long-term interaction scenarios. In the bargaining literature, human experiments and analyses reveal that equal-split offers are accepted more frequently and quickly than slightly unequal ones, in both one-shot and repeated interactions. We will consider moving the related work from the appendix to the main text.
> >
> > > **Reviewer RVe3:** Because of their mistaken supposition that the sender's ability to commit _requires_ long-term interactions, they do not consider the one-shot version (setting $\alpha$, in the paper's notation) Bayesian persuasion in their experiments
> >
> > As discussed earlier, we agree with the reviewer’s point and will include this experiment.
> >
> > > **Reviewer RVe3:** At the moment these settings are examined side by side, but what should be the takeaways from the similarities and differences from the results in the two settings?
> >
> > The reviewer’s observation is accurate: the $\gamma$ setting in BG is not equivalent to that in BP. The BG setting follows the standard Rubinstein's alternating-offer model, whereas the BP scenario can be viewed as a variant. We will include a BG experiment where the agent's role is fixed to enable comparison.
> >
> > **Notation**
> >
> > > **Reviewer RVe3:** If the receiver has no private information, there is no need to talk about this.
> >
> > We agree with the reviewer and will remove this discussion. Our initial intent was to explain the current literature, but this now appears unnecessary.
> >
> > > **Reviewer RVe3:** In footnote 1, it seems as though $q$ and $\phi$ are the wrong way round.
> >
> > We will remove this discussion.
> >
> > > **Reviewer RVe3:** The set of signalling schemes is denoted as blackboard bold $\psi$ but the signalling schemes themselves are elsewhere denoted as either $\phi$ or $\varphi$. Later $\psi$ is what determines the payoffs of the agents as a function of their commitments.
> >
> > Agreed, we will revise this section as per the reviewer’s suggestion.
> >
> > > **Reviewer RVe3:** Why do some symbols have tildes and others not?
> >
> > Agreed. Our intention was to use tilde notation to represent policies without awareness of the game structure and non-tilde notation for policies with such awareness. For instance, the title of Algorithm 2 does not include tilde. However, since the non-tilde notation is rarely used, we now opt to simplify this notation.
> >
> > > **Reviewer RVe3:** In Definition 3.3 $\tilde{\varphi}^*$ and $\tilde{\pi}^*$ have not been introduced.
> >
> > Agreed. There should not be an asterisk symbol here, and we have corrected this.
> >
> > **Language**
> >
> > > **Reviewer RVe3:** What are "cognitive misers"?
> >
> > *Cognitive misers* is a term in psychology that describes the human tendency to conserve cognitive energy by relying on mental shortcuts, heuristics, or simple rules of thumb instead of engaging in extensive deliberation or reasoning. This well-established concept highlights how individuals often prioritize efficiency over accuracy in decision-making and problem-solving. It is not a term we invented.
> >
> > > **Reviewer RVe3:** What does it mean for "solutions to become conceivable"?
> >
> > We will delete this sentence.
> >
> > > **Reviewer RVe3:**  The sentence after Assumption 3.2 was unclear to me (in part because the example with the coins had not yet been introduced).
> >
> > We will move Example 4.4 to the preliminaries section.
> >
> > > **Reviewer RVe3:** I think I understand what the authors mean here, but I would change the phrasing about different game structures...
> >
> > Thank you for the suggestion; we have made the changes accordingly.

---

> ### Comment · Reviewer_RVe3 · 2024-11-24
> **Reply To Authors**
>
> I thank the authors for their detailed reply to my comments, and I appreciate their taking on board of some of my suggestions to improve their paper. Below I offer some additional quick thoughts:
>
> - The analogy between BP and the ultimatum game is a useful one, and perhaps one that the authors should include in their paper to make their connection between bargaining and BP clearer.
> - As I have already mentioned in my reply to the public comment further above about "Claims", however, it does not seem as though the authors have fully internalised my main critique, or that they are especially willing to weaken the framing of their primary claim (which I view to be misleading).
> - I am glad that the authors will add BP experiments for the $\alpha$ setting, though as the manuscript has not been updated in this regard, I am unable to assess the validity and significance of these experiments.
> - I appreciate the authors' clarification that the main aim of their experiments is use LLMs as a proxy for humans in an attempt to investigate whether humans use the bargaining-esque strategies they consider in BP settings. However given that this is the authors' main aim I'm not convinced that the experiments they run are especially conclusive or that they advance the state of our knowledge beyond, say, that established by papers such as ["Simulating Strategic Reasoning: Comparing the Ability of Single LLMs and Multi-Agent Systems to Replicate Human Behavior"](https://arxiv.org/abs/2402.08189)  by Sreedhar & Chilton (2024) or ["Measuring Bargaining Abilities of LLMs: A Benchmark and A Buyer-Enhancement Method"](https://arxiv.org/abs/2402.15813) by Xin et al. (2024). In other words, based on the experimental results section (4.3) I don't think the authors sufficiently explore the aspects of the BP setting specifically (where commitments are about information revelation instead of setting prices) as opposed to bargaining more generally.
>
> In summary, my view of the paper continues to be that there are some interesting ideas in here, but that once the authors actually remove/revise the aspects of the paper that I view as misleading, the significance of the work will be greatly diminished and a substantial amount of effort will be required in order to bring it up to the standard that I expect from papers published at ICLR. As such, I unfortunately cannot recommend acceptance in its current state.

---

> > ### Author Response · Authors · 2024-11-25
> >
> > Thank you for your continued attention. Here we address the remaining issues:
> >
> > - Regarding points 1 and 2: We have replied them in the public comments. Please kindly refer to that.
> > - Regarding point 3: Due to time constraints and competing priorities, we will conduct this and other experiments later. Please consider this another "commitment" we propose :)
> > - Regarding point 4: We are not entirely sure what specific issue you are referring to here. Could you provide more detailed clarification? Alternatively, we would like to know what kind of experimental results would be considered conclusive. We are actively looking forward to opportunities to enhance the soundness of our experiments.

---

> > > ### Comment · Reviewer_RVe3 · 2024-11-27
> > > **Clarification of Previous Comment**
> > >
> > > Thanks for the reply, I agree that my fourth point could have been clearer. What I mean is that when I looked at, say the `sender_proposal_user_prompt` in Appendix E.4.5, the problem for the sender is framed essentially in bargaining terms due to the explicit formulation of the payoffs in terms of its action and the fact that the system prompt flags the possibility of retribution on behalf of the receiver. Thus, the idea of information revelation particularly seems to be mostly decorative and not that critical to the results – the sender is just instructed to choose a value of $\eta$ – and so represents less of an advance over existing works that consider the strategic and bargaining abilities of LLMs in simple games.

---

> > > > ### Author Response · Authors · 2024-11-30
> > > >
> > > > Thank you for the suggestions; we will revisit our experiments based on this perspective. However, we would like to point out that the purpose of this experiment is merely to observe the receiver's motivation and anti-exploitation behavior, and not to directly prove that BP is reducible to a bargaining game, as we have already provided a comprehensive discussion on this in the earlier sections.

---

### Official Review · Reviewer_giTx · 2024-11-02

**Soundness:** 3
**Presentation:** 3
**Contribution:** 2
**Rating:** 5
**Confidence:** 3

**Summary:**

The paper investigates connections between Bayesian persuasion and bargaining games. The key message as far as I'm concerned, is that the receiver may be able to gain more than in a standard solution to Bayesian persuasion by applying strategies that are more sophisticated than just best responding the the principal's commitment. The paper therefore proposes to apply the concept of bargaining game to Bayesian persuasion. It is shown that Bayesian persuasion with this new solution concept is reducible to bargaining games. Additionally, the authors conducted experiments with LLMs to test the theoretical claim.

**Strengths:**

The idea of studying the connection between persuasion and bargaining is very interesting. The paper has made a good effort to elaborate the background of the problem. The idea of using LLMs in the experiments is novel and inspiring. Overall I appreciate the endeavor of pushing research of Bayesian persuasion to more realistic setups beyond the standard theoretical model.

**Weaknesses:**

The main weakness of the paper as I see is a lack of clarity in its key message. I hope the authors could help to clarify the concerns in the rebuttal phase.

- The claim that Bayesian persuasion reduces to bargaining game is a bit confusing. It appears that what the authors actually mean is that a bargaining version of Bayesian persuasion, instead of Bayesian persuasion in its original form (where the sender commits), is reducible to bargaining games. If this is the case, the claim seems somewhat misleading, and in a sense it appears to me this claim will just apply to any kind of games, and it's not a property unique to Bayesian persuasion. It would be good if the authors could clarify these points.

- Following the above, it may be helpful to include some toy examples to illustrate the concepts of bargaining games, Bayesian persuasion in the bargaining setting, and standard Bayesian persuasion, as well as the differences between them.

- It may be helpful to include a bit more details about the experiment setup in the main paper, including who are the senders and receivers in the experiments, and how do they interact in the experiments. Tables 1 and 2 may be easier to parse if they are presented as plots.

**Questions:**

- As mentioned above, when you say Bayesian persuasion is reducible to bargaining game (e.g., in Lemma 3.5), do you mean by solving the bargaining game you also solve Bayesian persuasion in its original form (where the sender commits), or in a bargaining version of Bayesian persuaison? I suppose you mean the latter, as defined in Definition 3.4?

- Are there any properties unique to Bayesian persuasion that makes it reducible to bargaining games where other types of games (e.g., Stackelberg games) do not? Or is it the case that this claim applies to other types of games as well?

- Line 61, Page 2: The effect of history-dependent strategy is also discovered by Gan et al. (2023): Sequential principal-agent problem with communication: efficient learning and computation.

- The idea of the receiver's counter commitment agains the sender seems to be relevant to the folk theorems in game theory, as well as deception in Stacelberg games, where the follower, instead of responding optimally, commits to a different responding function to influence the equilibrium of the game: Birmpas et al (2020), Optimally deceiving a learning leader in Stackelberg games.

- The outcome of the bargaining game depends on the choice of the function $\omega_{BP}$. How does this influence the experiment results? Are the experiments (as well as the main goal of the paper) more about demonstrating that the receiver is able to gain more by using retaliatory strategies, so it's more from the receiver's perspective?

---

> ### Author Response · Authors · 2024-11-18
> **Official Comment (giTx - 1) by Authors**
>
> Thank you for your insightful comments. We have addressed the issues regarding the weaknesses section in our public response, as your views are similar to those of reviewer RVe3. Here, we will proceed to answer the other questions.
>
> > **Reviewer giTx:** It appears that what the authors actually mean is that a bargaining version of Bayesian persuasion, instead of Bayesian persuasion in its original form (where the sender commits), is reducible to bargaining games... It may be helpful to include a bit more details about the experiment setup in the main paper
>
> The detailed discussion is in Public Response (1) and (2). Please kindly refer to that.
>
> > **Reviewer giTx:** Following the above, it may be helpful to include some toy examples to illustrate the concepts of bargaining games, Bayesian persuasion in the bargaining setting, and standard Bayesian persuasion, as well as the differences between them.
>
> See example 4.1, Grading Students, which is a classical Bayesian persuasion task. The detailed analysis is in appendix D. Based on the assumptions of BP, the sender proposes a signaling scheme:
> - For instance, the sender commits to recommending a good student with 100% probability and a bad student with 50% probability. We consider this effectively proposing a reward assignment, which the receiver can observe. If the receiver best responds to this signaling scheme, the sender will receive an expected reward of $2/3$, and the receiver will receive an expected reward of $0$.
> - If the sender becomes more honest, for example, recommending a good student with a probability of $1$ and a bad student with a probability of $25\%$, then when the receiver best responds, the sender and receiver will receive expected rewards of $1/2$ and $1/6$, respectively.
> - In both cases, the receiver can also choose to always ignore the sender. In this case, their expected rewards will both be $0$.
>
> Thus, it can be seen that Grading Students is identical to the bargaining game (Example 4.3), except that the bargaining game directly proposes an offer regarding the reward assignment, while bp achieves this by committing to a signaling scheme.
>
> **Other Issues**
>
> > **Reviewer giTx:** Tables 1 and 2 may be easier to parse if they are presented as plots.
>
> We will turn table1 and table2 into plots. This is an excellent suggestion.
>
> > **Reviewer giTx:** Do you mean by solving the bargaining game you also solve Bayesian persuasion in its original form (where the sender commits), or in a bargaining version of Bayesian persuaison?
>
> We believe we are addressing the original form of Bayesian persuasion, and the specific insights have been explained in the public response.
>
> > **Reviewer giTx:** Are there any properties unique to Bayesian persuasion that makes it reducible to bargaining games where other types of games (e.g., Stackelberg games) do not?
>
> This is a great question. We believe the primary reason Bayesian persuasion can be viewed as a bargaining problem is that, in all cases, there is always a disagreement point (see definition 2.5), where the receiver simply best responds according to the prior distribution. This could be caused either by the receiver ignoring the sender or by the sender sending uninformative signals.
>
> > **Reviewer giTx:** The effect of history-dependent strategy is also discovered by Gan et al. (2023)
>
> We are aware of this team's work, and we have already cited some of their papers in appendix B.2. One of their earlier works on sequential persuasion was from 2022. However, we have now added a citation to their 2023 paper as well.
>
> > **Reviewer giTx:** The idea of the receiver's counter commitment agains the sender seems to be relevant to the folk theorems in game theory, as well as deception in Stacelberg games
>
> The studies mentioned by the reviewer are indeed very relevant. We will discuss it in the manuscript.
>
> > **Reviewer giTx:** The outcome of the bargaining game depends on the choice of the function ωBP. How does this influence the experiment results?
>
> Our experiments do not require considering the choice of solution concepts (e.g. Nash bargaining solution or Kalai–Smorodinsky bargaining solution). The detailed discussion is in Public Response (2). Please kindly refer to that.

---

### Official Review · Reviewer_9d3n · 2024-11-04

**Soundness:** 2
**Presentation:** 2
**Contribution:** 2
**Rating:** 3
**Confidence:** 3

**Summary:**

This paper provides a new perspective on Bayesian persuasion by showing it can be reduced to a bargaining game.
The authors tries to show that receivers can develop anti-exploitation strategies when they're aware of the game structure. By simulations with large language models, the authors show that these bargaining dynamics emerge naturally in persuasion tasks.

**Strengths:**

1. The theory looks solid and the perspective to compare a BP game to a bargain game is very interesting.

**Weaknesses:**

1. The authors claimed the reason for LLM agents as the person to test the mechanism is “It is immediate  for LLMs to provide their interpretation as to why such decisions are made by giving appropriate prompts.” Caution must be taken since LLM tend to rectify the decision it just made. This is dubious as a recent paper shows https://arxiv.org/abs/2410.19599.
2. Even the LLM simulation setting is not clearly described. Context like when will the game end or how many times do these LLM agent play, are these repeated or started from afresh, or the temperature of the models are not described at all.
3. How does the results of LLM game compared to the one with real human? In Table 1 and 2 summarize the results of LLM agents playing the task games. And these tables are not mentioned in the draft any more, which adding more confusing to the audience. What does the experiment result try to establish? The display is very confusing.
4. If the author want to demonstrate how the receiver in a BP task can use an action rule with game structure awareness to achieve higher expected gain, the statistical analysis are required. In the discussion follows, the author seems to describe the similarity in strategic behavior.

**Questions:**

1. I suggest the authors look into the paper https://arxiv.org/abs/2410.19599 and argue why the LLM simulation can provide any insights into the their scenario.
2. More clarification of the simulation context is required
3. More explanation of Table 1 and Table 2 and the statistical analysis is encouraged.

---

> ### Author Response · Authors · 2024-11-12
> **Mis-paste review**
>
> Dear Reviewer, Thank you for reviewing our submission. We believe that the review is mis-pasted from some other submissions. Could you please replace the review with the intended one? Thank you

---

> ### Author Response · Authors · 2024-11-18
> **Official Comment (9d3n - 1) by Authors**
>
> Thank you for your attention to the role of LLMs in our work. Regarding the experimental section, we have addressed it in Public Response (2). We believe this addresses the main concerns in Weakness 1 (also Q1, see **Experimental Motivation**), Weakness 2 (also Q2, see **Experimental Setup**), and Weakness 4 (also Q3, see **Experimental Motivation**).
>
> **Weakness 1 and Q1**
>
> > **Reviewer 9d3n:** I suggest the authors look into the paper [https://arxiv.org/abs/2410.19599](https://arxiv.org/abs/2410.19599) and argue why the LLM simulation can provide any insights into the their scenario.
>
> First, we want to highlight that the ICLR submission deadline occurred nearly a month before the preprint article mentioned by the reviewer was posted on arXiv. Thus, we had not considered this potential risk earlier. The preprint cited by the reviewer is indeed intriguing and discusses testing LLMs in 11–20 Money Request Games, concluding with the claim that "LLMs Rarely Mimic Human Behavior" (which is the title of one of its sections).
>
> We believe that LLMs can still serve as a human proxy in our context. As stated in the **Experimental Motivation** section of Public Response (2), the results of LLMs in bargaining games align with those from human experiments.
>
> Furthermore, there are many peer-reviewed articles that leverage LLMs to study social, behavioral, and cognitive sciences, which seem to support views contrary to those presented in the preprint mentioned by the reviewer:
> 1. Meng, Juanjuan. "AI emerges as the frontier in behavioral science." *Proceedings of the National Academy of Sciences* 121.10 (2024): e2401336121.
> 2. Mei, Qiaozhu, et al. "A Turing test of whether AI chatbots are behaviorally similar to humans." *Proceedings of the National Academy of Sciences* 121.9 (2024): e2313925121.
> 3. Li, Yuxiao, et al. "The Geometry of Concepts: Sparse Autoencoder Feature Structure." *arXiv preprint arXiv:2410.19750* (2024).
>
> This is a very intriguing perspective to us, and we will include a discussion of all these papers (including the one mentioned by the reviewer) in the manuscript.
>
> **Weakness 2 and Q2**
>
> > **Reviewer 9d3n:** 1. Even the LLM simulation setting is not clearly described. Context like when will the game end or how many times do these LLM agent play, are these repeated or started from afresh, or the temperature of the models are not described at all.
>
> We have supplemented additional experimental details in the **Experimental Setup** section of Public Response (2). Please refer to that section.
>
> In the manuscript, we describe the game termination setting on line 429:
> "($\gamma$) The agents play the game multiple times. The maximum time $T$ is sampled by a memoryless distribution, which introduces the shadow of the future (Bo, 2005), preventing players from predicting when the game will end. We defer the exact prompt used to Appendix E."
>
> Experimental repetitions are equivalent to restarting the game, but the agents are aware of the results from previous games, making their strategies history-dependent. This is consistent with the setup of repeated games in game theory.
>
> While we did not detail the temperature and other LLM parameter settings in the manuscript, we plan to include these in a future version. (However, in the initially submitted supplementary materials, we provided the full code (including all parameters and prompts) as well as the outputs of all experiments.)
>
> **Weakness 3**
>
> > **Reviewer 9d3n:** How does the results of LLM game compared to the one with real human? ...What does the experiment result try to establish?
>
> It is important to note that our paper is not about evaluating the capabilities of LLMs, which is the primary focus of the preprint mentioned by the reviewer. Our experiments do not include any human studies; all experiments are conducted using LLMs. In this context, LLMs are utilized as tools or solvers to study the properties of two specific problems, particularly focusing on agent behavior in these problems.
>
> Our claim that LLMs can serve as human proxies stems from the observation that LLMs produce similar results in bargaining scenarios. Additionally, when analyzing the motivations behind their strategies, we found evidence of retaliatory thinking, and similar strategies emerged in Bayesian persuasion. Our core focus, as aligned with our title, is that Bayesian persuasion is a bargaining game.
>
> **Weakness 4 and Q3**
>
> > **Reviewer 9d3n:** 1. If the author want to demonstrate how the receiver in a BP task can use an action rule with game structure awareness to achieve higher expected gain, the statistical analysis are required. In the discussion follows, the author seems to describe the similarity in strategic behavior.
>
> The reviewer’s observation is correct that we focus on the strategies of LLM players. The reasons for this are discussed in the **Experimental Motivation** section of Public Response (2). Please kindly refer to that.

---

### Author Response · Authors · 2024-11-18
**Public Response (1): Claim**

The core opinions of reviewers RVe3 and giTx are essentially the same: when we introduce an anti-exploitation commitment for the receiver, the game becomes a variant of Bayesian persuasion (BP) rather than BP itself, which has caused some confusion.

It is notable that, in our work, the receiver's behavior differs from that in classical BP. In classical BP, the analysis assumes that the receiver uses the Bayesian decision rule to infer the state based on the signal and then selects the best posterior action. This is considered the best response to the signaling scheme. When the receiver behaves this way and the sender's signaling scheme satisfies the obedience constraint (the constraint in Equation 2), the receiver fully follows the sender's recommendation. Thus, in most literature, when a signaling scheme satisfies obedience, the receiver's behavior is assumed to comply.

From this perspective, we consider **the receiver in BP to have a very naive unconditional commitment,** i.e., the receiver commits to adopting the above behavior (Bayesian decision rule) regardless of the sender's signaling scheme. The corresponding logic applies to the bargaining game as well: the responder's best response to the reward assignment proposed by the proposer is to accept it if the reward assigned to the receiver exceeds the disagreement point (which in BP corresponds to satisfying obedience). This means that **the standard behavior in BP leads to the result being a standard subgame perfect equilibrium of an ultimatum game.**

However, we know this is not how humans behave in experiments, and if the game were to be played repeatedly, the agents' behavior would also be different. The specific background of the bargaining game is provided in Appendix B.1 of the manuscript. Therefore, this work focuses on a general solution concept for bargaining games (Definition 2.6), which establishes a rule for reaching an agreement in all bargaining game settings (line 211), such as the Nash bargaining solution. For example, this solution suggests that both parties should split the rewards equally (e.g., in Example 4.4, each party receives 50 coins).

One may think that the BP problem then remains the optimization problem in Equation 2 but with the constraint replaced by one related to the receiver satisfaction check. This shows that the original obedience constraint is a special case. However, this is still insufficient, as it implies the receiver must first commit to a satisfaction function for the sender. **This is equivalent to the receiver proposing a reward assignment that the sender can either accept or reject.** (if rejected, the sender can choose not to send any meaningful information, e.g., sending the same signal for all states, as described in line 263). The receiver becomes the proposer and the sender becomes the responder.

Thus, the BP solution is defined as a joint commitment (Definition 3.3). Based on Assumption 3.2, we know there exists at least one outcome that is Pareto superior to the disagreement point; otherwise, the sender would have no motivation to persuade the receiver, and the receiver would have not motivation to follow the sender's recommendations.

Assume there is a dynamic function $f: \tilde{\varphi}\times \tilde{\pi} \to \tilde{\varphi}\times \tilde{\pi}$, which allows both parties to modify their policies based on the current circumstance. The BP solution is the fixed point of this dynamic function, where **both parties are satisfied with the reward assignment, and no one wishes to unilaterally set the reward assignment to the disagreement point.** We argue that only this definition can clearly resolve the BP problem. The logic of this aligns with the development of the bargaining game.

The above analysis focuses on the final outcome without specifying how to reach it. We use a variant of Rubinstein's alternating-offer model from the bargaining game, with the experimental setup detailed in Appendix E and some results (terminal outputs) in Appendix F.

**The claim of our work should be that:** the classic Bayesian persuasion is reducible to a bargaining game, and its solution is not the sender's obedient signaling scheme that optimizes their own reward in the classical analysis, but rather an agreement on a reward assignment achieved through the policies of both parties. To find this solution, we equivalently solve a joint commitment. Our reduction (Lemma 3.5) specifies this: solving BP requires a solution to a bargaining game, and once a solution to the bargaining game is found, we can solve the BP problem in polynomial time, as it is merely a straightforward correspondence (an optimization problem with only constraint conditions).

---

> ### Comment · Reviewer_RVe3 · 2024-11-24
> **The Authors' Claims Remain Incorrect**
>
> I will reply fully to the authors' response to my own review below shortly. For now I simply wish to point out that even in their response above the authors repeat their incorrect claim about what their paper shows:
>
> > **The claim of our work should be that:** the classic Bayesian persuasion is reducible to a bargaining game
>
> As I wrote in my review:
>
> > what the authors _actually_ show is that a variant of Bayesian persuasion where the receiver is also given the ability to commit and the agents are able to interact beforehand is reducible to a bargaining game
>
> I will also note that their interpretation that "the receiver in BP [makes] a very naive unconditional commitment" is not really in keeping with the classic Bayesian Persuasion model, where the point is that the receiver **has no such commitment ability**. The authors can certainly claim that not giving the receiver some kind of commitment ability in the BP model is a shortcoming that prevents that model from capturing settings that they are interested in, but they cannot claim that the receivers in the classic BP model are being systematically naive by not making use of an ability (commitment) **that they are not assumed to have** in that model.

---

> > ### Author Response · Authors · 2024-11-25
> > **Reaffirming Our Claim with Further Clarifications (a)**
> >
> > To clarify, we want to reiterate our claim explicitly: The classic Bayesian persuasion is reducible to a bargaining game. The disagreement between us and reviewer RVe3 arises from the definition of classic Bayesian persuasion (BP) and the ambiguity in the discussion of the **problem setting** and **solution**. We will restate our argument in the most direct way and hope the reviewer can reconsider our work with this perspective.
> >
> > Let us first focus on what BP represents as a problem. It is important to note that we follow the definition provided by the original author of Bayesian persuasion: [Kamenica, Emir. "Bayesian persuasion and information design." _Annual Review of Economics_ 11.1 (2019): 249-272.](https://www.kevindorst.com/uploads/8/8/1/7/88177244/kamenica_-_bayesian_persuasion_and_information_design_-_2019.pdf) In Section 2.1 "The Basic Model," the timing of the game is described as follows, which we express in our notation:
> >
> > 1. The sender chooses a signaling scheme $\varphi$.
> > 2. The receiver observes which signaling scheme was chosen.
> > 3. Nature chooses $s$ according to $\mu_0$.
> > 4. Nature chooses $\sigma$ according to $\varphi(s)$.
> > 5. The receiver observes the realized $\sigma$.
> > 6. The receiver takes action $a$.
> >
> > And our problem setting is defined as follows:
> >
> > 1. The sender chooses a signaling scheme $\varphi$.
> > 2. The receiver observes which signaling scheme was chosen **and decides an action rule $\pi$**.
> > 3. Nature chooses $s$ according to $\mu_0$.
> > 4. Nature chooses $\sigma$ according to $\varphi(s)$.
> > 5. The receiver observes the realized $\sigma$.
> > 6. The receiver takes action $a$ **according to its action rule $\pi$**.
> >
> > This is completely equivalent to:
> >
> > 1. The sender chooses a signaling scheme $\varphi$.
> > 2. The receiver observes which signaling scheme was chosen.
> > 3. Nature chooses $s$ according to $\mu_0$.
> > 4. Nature chooses $\sigma$ according to $\varphi(s)$.
> > 5. The receiver observes the realized $\sigma$.
> > 6. The receiver **decides an action rule $\pi$** and takes action $a$ **according to its action rule $\pi$**.
> >
> > There is no conflict with the original definition. Therefore, our claim remains accurate: we analyze Bayesian persuasion itself, not a variant of it. This shows that the receiver does not need any commitment regarding its satisfaction function to complete the game.
> >
> > ---
> >
> > We now discuss the **solution** perspective, where commitment becomes relevant.
> >
> > First we argue that **even in the classic Bayesian persuasion, the receiver inherently has an implicit commitment.** As mentioned in our rebuttal, while the reviewer believes this is incorrect, we argue otherwise. Understanding this implicit commitment is key to grasping our contribution:
> >
> > - **This implicit commitment refers to the receiver's naive rationality.** Specifically, it means the receiver will choose actions that maximize their posterior expected reward.
> > - This is common knowledge to both players and occurs before the sender designs and commits to a signaling scheme. In the classic analysis, the sender’s decisions heavily depend on this, which is why we referred to it as a "commitment."
> > - Without this fundamental implicit commitment, the sender cannot use an incentive-compatible (or obedience) signaling scheme to persuade the receiver. **Because in this case the sender does not know under what conditions the receiver would follow their recommendation** or to what extent the receiver is satisfied.
> >
> > Discussions of economic rationality being common knowledge are abundant in economic studies and are explicitly mentioned in BP. Here, we clarify that this can be considered a form of implicit commitment.

---

> > > ### Comment · Reviewer_RVe3 · 2024-11-27
> > > **I Continue To Disagree, But Less Than Before**
> > >
> > > I thank the authors for the effort in carefully spelling out their claims, and I believe this has reduced some of the confusion and concern regarding what they are claiming. Still, I think the wording of their claims diverges from standard interpretations of the word "commitment" in game theory that makes the core claim of their paper (including the title!) misleading, as I explain below.
> > >
> > > **Problem Setting**
> > >
> > > Thank you for clarifying this. I agree that you that the description of the problem scenario here is just the classic BP setting. The authors should correct me if I am wrong, but my current interpretation is therefore that the authors are considering a different _solution concept_ as opposed to a different game. For example, one can consider the Stackelberg equilibria or the Nash equilibria of a normal-form game. The former implicitly introduces a kind of a additional structure where one player gains a commitment ability and the game is split into two stages, but it is still normal to talk about the Stackelberg equilibria _of the original game_. As far as I am concerned, this is a perfectly ok thing to do. (Note however, that this does not mean that "Stackelberg games are Normal-Form Games".)
> > >
> > > **Solution Concept**
> > >
> > > This is where I continue to disagree.
> > >
> > > - First, I disagree with the conceptualisation of the receiver "implicitly committing". For example, imagine a simple extensive-form game with perfect information where the first player chooses between two actions and then the second player chooses between two actions. We do not say that the second player "implicitly commits" to anything when selecting their strategy. The idea of commitment in games is well-theorised at this point and I believe that the authors' departure from conventional usage of this word will be misleading to relevant audiences. This is especially true because in classic Bayesian Persuasion there is a very real, concrete, conventional sense in which the sender commits that does not apply to the receiver.
> > > - Second, even if this were the case, I disagree that it is rational for the receiver to "commit" (in the authors' usage: select a strategy) to playing subgame imperfectly (at least not without additional assumptions, as I remark later). Indeed, the motivation of subgame imperfection is that it rules out non-credible threats, which are deemed _irrational_. The only thing that makes such commitments rational is if they are actually credible (e.g., because one can punish the other player in future rounds if they don't respond appropriately to the commitment, or because one has a legible way of committing, like throwing one's steering wheel out of the window in a game of chicken).
> > > - Third, this notion of implicit commitment is not required for the sender to know what the receiver would do, only (common) knowledge of rationality and payoffs. For example, (under conventional usage of the word "commitment") in Nash equilibrium there need be no notion of implicit commitment, rather I am able to reason about what you would do as a rational actor given your payoffs (where the Nash equilibrium solution concept defines what "rational" means in this setting).
> > >
> > > So in sum, the authors claim seems to be something like: "if we take a classic [Bayesian persuasion] game and consider solutions in which bargaining is allowed via commitments made by the receiver [in addition to the commitments made by the sender, which are already part of the classic BP formalism], then we can view this as bargaining game". But this is neither surprising nor interesting by itself. I can say this about any game I like where one player has a commitment ability and then I introduce some commitment ability for the other player as well, the fact that it is a Bayesian persuasion game in particular does not make a difference in this regard.
> > >
> > > To really try to get this point across: I cannot simply claim that "Stackelberg games are bargaining games" or that "in Stackelberg games the follower also implicitly commits to their strategy" – that's simply not true. I _can_ claim that "we should re-characterise Stackelberg games as bargaining games" or "most things we analyse as Stackelberg games should actually be analysed as bargaining games" or "if the Stackelberg game is temporally extended or some other variation is made then we can analyse this as a bargaining game", but that is not the sort of claim that the authors actually make.

---

> > > > ### Author Response · Authors · 2024-11-30
> > > >
> > > > Thank you for the continued attention.
> > > >
> > > > **Problem Setting**
> > > >
> > > > We are glad to have reached a consensus here. If your example is simply an analogy (i.e., if you are not claiming that BP is a Stackelberg game and a bargaining game is a normal-form game), then your interpretation is correct.
> > > >
> > > > We are considering changing the title to "Bayesian Persuasion Is Reducible to a Bargaining Game," as this phrasing is more precise. The reduction specifically addresses the solution; that is, if a solution to the bargaining game is found, then the solution to Bayesian persuasion can also be determined. In other words, we are revisiting Bayesian persuasion from the perspective of a bargaining game.
> > > >
> > > > ---
> > > >
> > > > **Solution Concept**
> > > >
> > > > **The First Issue**
> > > > We understand that "commitment" is a well-theorized concept, and we are using it deliberately.
> > > >
> > > > > Reviewer: For example, imagine a simple extensive-form game with perfect information where the first player chooses between two actions and then the second player chooses between two actions. We do not say that the second player "implicitly commits" to anything when selecting their strategy... This is especially true because in classic Bayesian Persuasion there is a very real, concrete, conventional sense in which the sender commits that does not apply to the receiver.
> > > >
> > > > For the example provided, the second player does indeed commit to acting rationally. Rationality is a common and widely accepted assumption, and we consider it a form of commitment.
> > > >
> > > > To clarify, the receiver's rationality in BP can be seen as a commitment. The receiver commits to a function: given a $\mu_0$ and $r^j$, for any possible signaling scheme $\varphi$, upon receiving signal $\sigma$, they choose $a = \arg\max_{a} \sum\limits_{s}\mu(s\mid \sigma)r^j(s,a)$, where $\mu(s\mid\sigma) = \frac{\mu_0(s)\varphi(\sigma\mid s)}{\sum_{s_i}\mu_0(s_i)\varphi(\sigma\mid s_i)}.$
> > > >
> > > > Just as the sender commits to their signaling scheme before the game starts, the receiver also commits to this function and adheres to it during the game. We do not understand why the reviewer believes this is not a form of commitment.
> > > >
> > > > We argue that the receiver in Bayesian persuasion indeed has a commitment (rationality assumption). In most research on BP, the receiver's behavior is fixed as choosing the posterior-optimal action, and the focus is on optimizing the sender's signaling scheme. Once the sender's optimization is complete, their discussions end. Our assertion that the receiver has a commitment is meant to highlight that past studies have overlooked this aspect and its potential as a bargaining problem.
> > > >
> > > > ---
> > > >
> > > > **The Second Issue**
> > > >
> > > > > Reviewer: Even if this were the case, I disagree that it is rational for the receiver to "commit."
> > > >
> > > > As noted above, the receiver's behavior exemplifies rationality. Nearly all Bayesian persuasion papers analyze this way; this is not our claim. The reviewer can refer to any of the papers we previously cited. Additionally, even in axiomatic models of bargaining games, the rationality assumption is common.

---

> > > > ### Author Response · Authors · 2024-11-30
> > > >
> > > > **The Third Issue**
> > > >
> > > > > Reviewer: this notion of implicit commitment is not required for the sender to know what the receiver would do.
> > > >
> > > > It is required. The sender must know that the receiver is rational for the subsequent analysis. It is inconceivable to persuade an irrational receiver whose behavior is unpredictable. And it is the obedience constraint of the signaling scheme that characterizes this concern.
> > > >
> > > > > Reviewer: the authors’ claim seems to be something like... But this is neither surprising nor interesting by itself... To really try to get this point across: I cannot simply claim that "Stackelberg games are bargaining games" or that "in Stackelberg games the follower also implicitly commits to their strategy."
> > > >
> > > > This concern aligns with the issue raised by Reviewer giTx: What makes BP special, and can our conclusions be extended to other Stackelberg games?
> > > >
> > > > Our response is that the most critical property is the existence of a clear disagreement point in communication: the receiver acts based only on the prior. This can arise either because the receiver does not respect the sender’s signaling scheme or because the sender issues an uninformative signal (resulting in the receiver’s posterior equaling the prior). Furthermore, this disagreement point is an undesirable outcome. Otherwise, there would be no need for communication; the sender would have no influence over the receiver’s behavior.
> > > >
> > > > Our focus is solely on Bayesian persuasion. Other Stackelberg games may not share this property. Nevertheless, BP remains a broad concept, as BP = Cheap Talk + Sender's Commitment (this is a community-accepted view, not our claim).
> > > >
> > > > At this stage, the evaluation of the novelty becomes subjective to the domain. To argue this point inevitably risks sounding self-congratulatory, but we would like to provide some background on Bayesian persuasion to shift the reviewer’s perspective.
> > > >
> > > > Currently, there are tons of studies focusing on long-term interactions in Bayesian persuasion. This property is particularly crucial when the players are learning algorithms. The disagreement point is a highly “attractive” equilibrium. Without careful consideration, it is easy to fall into this equilibrium and remain stuck, which is a terrible outcome. Agents must possess some form of “tolerance” to learn effectively. We believe the community has not yet realized this. Many current methods are, in fact, special cases of strategies under bargaining games. We hypothesize that without explicitly pointing out that "Bayesian persuasion is reducible to a bargaining game," a significant number of similar papers may continue to emerge.
> > > >
> > > > ---
> > > >
> > > > > Reviewer: I _can_ claim that "we should re-characterize Stackelberg games as bargaining games" or "most things we analyze as Stackelberg games should actually be analyzed as bargaining games" or "if the Stackelberg game is temporally extended or some other variation is made then we can analyze this as a bargaining game," but that is not the sort of claim that the authors actually make.
> > > >
> > > > We sincerely appreciate the reviewer’s insightful thoughts. These three points effectively summarize our intended message. Please refer to our previous discussion for details. However, we will replace "Stackelberg games" in this phrasing with "Bayesian persuasion."

---

> > ### Author Response · Authors · 2024-11-25
> > **Reaffirming Our Claim with Further Clarifications (b)**
> >
> > Specifically, the classic analysis of BP proceeds as follows:
> >
> > 1. The receiver declares that they will choose actions that maximize their posterior expected reward, regardless of the sender’s signaling scheme, and the sender is aware of this.
> > 2. The sender chooses a signaling scheme $\varphi$ (which satisfies incentive compatibility).
> > 3. The receiver observes which signaling scheme was chosen.
> > 4. Nature chooses $s$ according to $\mu_0$.
> > 5. Nature chooses $\sigma$ according to $\varphi(s)$.
> > 6. The receiver observes the realized $\sigma$ and updates their posterior belief.
> > 7. The receiver takes action $a$ that maximizes their posterior expected reward.
> >
> > This shows that the receiver’s implicit commitment is a special case, equivalent to revealing their naive satisfaction function before the sender’s decision-making (discussed in Section 3.4 of our work).
> >
> > To draw a comparison, consider the ultimatum game. In Example 4.4, two players split 100 coins. The subgame perfect equilibrium is analyzed as follows:
> >
> > 1. The responder declares that they are rational: they have two actions, A and B, and will choose A if its expected reward exceeds that of B. The proposer knows this.
> > 2. The proposer suggests a split that the responder would accept and that maximizes the proposer’s own reward. In this case, they suggest (99, 1).
> > 3. The responder either accepts or rejects the offer. In this case, they accept, since accepting yields a reward of 1, whereas rejecting yields 0.
> >
> > The analyses are entirely equivalent. In the classic analysis of BP, the first three steps represent a bargaining process, aligning exactly with the subgame perfect equilibrium (SPE) of a bargaining game. **The sender suggests a reward assignment through a signaling scheme.** Steps 4–7 in BP reflect the realization of the signals. This demonstrates that the analysis of BP is a special case of bargaining game analysis.
> >
> > However, this classic analysis may be insufficient from the perspective of bargaining games.
> >
> > For the receiver, a straightforward strategy would be to become greedier by committing to an alternative satisfaction function, aiming to extract more information from the sender. However, this leads to a different issue: the receiver essentially proposes a reward assignment by committing to various satisfaction functions. At this point, they act as the proposer in the bargaining game, as analyzed in Appendix C.2. This phenomenon is referred to as **The Power of Commitment.**
> >
> > ---
> >
> > To address the question: When can we claim to have solved a bargaining game? The answer lies in ensuring that the outcome does not fall to the disagreement point, as this point is Pareto dominated by any agreement. Various solution concepts exist in bargaining game studies, each corresponding to a specific agreement.
> >
> > Returning to Bayesian persuasion, the implicit commitment of the receiver (economic rationality) ensures clarity in how agreements are reached. However, this is merely one solution concept. Thus, we reexamined BP through the lens of bargaining game development.
> >
> > We first address a key question: **When can we claim to have solved Bayesian persuasion?** We assume that any agreement between the sender and receiver must be Pareto optimal relative to the default disagreement, as otherwise, the communication would be unnecessary. **The solution should characterize what constitutes reaching an agreement.** This characterization excludes any game dynamics and can be completed before actual signal realization (steps 3–6 in the classic BP). This characterization may have led the reviewer to mistakenly believe we were discussing a variant of Bayesian persuasion rather than Bayesian persuasion itself.
> >
> > ---
> >
> > Our contribution lies in framing Bayesian persuasion within the bargaining game framework, emphasizing the necessity of joint commitment and bargaining solution concepts in persuasion. Additionally, bargaining games can exhibit more complex strategic behavior in long-term interactions. Viewing BP as a bargaining game also explains many existing studies well (Appendix B.2).
> >
> > The misunderstanding likely stems from writing issues. But we do not intend to weaken our main claim (though we will soften discussions about long-term requirements). The "curse of knowledge" may have led us to overlook points that should have been emphasized earlier. We appreciate the reviewer’s questions, which brought this to our attention. **We will revise the manuscript and incorporate all the points discussed in this rebuttal.**

---

> ### Comment · Reviewer_giTx · 2024-11-30
>
> I went through this long thread of discussion and I mostly agree with Reviewer RVe3. The paper looked at a model that has the same structure as Bayesian persuasion but uses a different solution concept. In the solution concept, the receiver has a kind of "pre-commitment" power and can pre-commit to a response rule that is different from best responses in BP. While the observations are interesting and hold some values for understanding BP, it appears that the correct message should be: BP with this new solution concept is reducible to bargaining games. The current presentation reads more like BP in its standard form is a special case of bargaining game.
>
> I think the observations the authors made are common in other types of Stackelberg games too. There have indeed been papers exploring similar dynamics where the follower in a Stackelberg game commits to a response rule different from the best response (see e.g., Imitative follower deception in stackelberg games. Gan, J., Xu, H., Guo, Q., Tran-Thanh, L., Rabinovich, Z., & Wooldridge, M.). The current paper seems to align more closely with this line of research (while showing a novel connection to bargaining games). The authors may consider revising the perspective from which their contributions are framed. Perhaps some questions to be asked along this line are: What do we get from the new solution concept, compared with the standard BP solution concept? Are they more appropriate for centain scenarios? How much does the receiver benefit from this new type of interaction? What are the new challenges involved (e.g., in terms of computating a solution/equilibrium)?
>
> Overall, I think the paper shows interesting observations but seems to have not framed its contributions in the most accessible way.

---

> > ### Author Response · Authors · 2024-11-30
> >
> > We sincerely appreciate all the reviewers' suggestions, constructive questions, and the time they have dedicated to reviewing our work. We still wish to continue clarifying the overall logic of our study, as the rebuttal process has provided us with new insights into this work, thanks to the contributions of all the reviewers.
> >
> > > Reviewer giTx: While the observations are interesting and hold some values for understanding BP, it appears that the correct message should be: BP with this new solution concept is reducible to bargaining games. The current presentation reads more like BP in its standard form is a special case of bargaining game.
> >
> > The points mentioned by the reviewer are not conflicting; we simply emphasized the latter (i.e., the second point below) in the rebuttal. Let us briefly explain the logic:
> > - Standard BP problem setting = bargaining game (ultimatum) problem setting
> > - Standard BP problem setting + standard BP solution concept = bargaining game (ultimatum) problem setting + subgame perfect equilibirum (a solution concept)
> > - Standard BP problem setting + standard BP sender behavior -> receiver may not be satisfied -> failed bargaining outcomes
> > - Standard BP problem setting + any bargaining solution concept -> solved
> >
> > The discussion about the receiver's commitment might be a significant distractor. Intuitively, we would explain it like this:
> > - Again: Standard BP problem setting + standard BP sender behavior -> receiver may not be satisfied -> failed bargaining outcomes -> unresolved
> > - How do we know under what circumstances the receiver is satisfied? (The sender should also be satisfied.) -> Given the current circumstance, they will not further modify their strategies.
> > - Then they will both commit to play their strategies.
> > - We know that: cheap talk + long-term interaction (or commitment devices) + sender's commitment = Bayesian persuasion
> > - Now we say that: cheap talk + long-term bargaining process (or commitment devices) + sender's commitment + receiver's commitment = our work
> > - Long-term bargaining process can be solved by bargaining solution concepts.
> > - Standard BP problem setting + any bargaining solution concept + joint commitment -> BP solved
> >
> > > Reviewer giTx: What do we get from the new solution concept, compared with the standard BP solution concept? Are they more appropriate for certain scenarios?
> >
> > Currently, we believe it can be understood as follows:
> > - We aim to use algorithms to guide practice. However, in bargaining problems, the standard solution concept implies that, in a scenario of dividing 100 coins, the receiver would only be given 1 coin. In numerous human experiments on bargaining games, such a scenario is usually rejected. Consequently, the sender would lose the reward that could have been achieved through communication (persuasion). Therefore, different solution concepts are necessary.
> > - This has meaningful implications for long-term persuasion scenarios. We point out that some existing works have already proposed strategies that are effectively related to bargaining.
> >
> > > Reviewer giTx: How much does the receiver benefit from this new type of interaction?
> >
> > This formulation not only accounts for changes in the receiver but also includes the sender's awareness of these changes in the receiver. In other words, even in a single interaction, the sender may allocate more benefits to the receiver out of fear of rejection. This is similar to the dynamics observed in bargaining games.
> >
> > > Reviewer giTx: What are the new challenges involved (e.g., in terms of computating a solution/equilibrium)?
> >
> > We have not proposed a learning method as a solver at this point. The current experiments rely on the zero-shot testing capability of LLMs. In fact, more sophisticated algorithms that are better suited for this scenario could be used, including those that consider different commitment devices and various learning methods (such as RL or LLM-based approaches).
> >
> > > Reviewer giTx: There have indeed been papers exploring similar dynamics where the follower in a Stackelberg game commits to a response rule different from the best response (see e.g., Imitative follower deception in stackelberg games. Gan, J., Xu, H., Guo, Q., Tran-Thanh, L., Rabinovich, Z., & Wooldridge, M.).
> >
> > Thanks for the recommendation. We took a preliminary look and found this to be a highly relevant topic. Although we haven’t fully digested it yet, we understand why the reviewer suggested this as a potential extension. We will add these discussions once we have thought it through.

---

### Author Response · Authors · 2024-11-18
**Public Response (2): Experiments (a)**

All three reviewers found our experiments confusing. We acknowledge that there were oversights in our writing, and we provide a unified response here.

**Experimental Motivation**

In Section 3, we explained what the solution to the bargaining problem (BP) should be and formalized it using a reduction. This shows that specifying a bargaining solution is sufficient to resolve the BP. Most bargaining solution concepts are axiomatic methods, such as the Nash bargaining solution. For instance, in Example 4.4, when dividing 100 coins, the agents would split it equally (the symmetry axiom). **However, existing bargaining solutions generally fail to accurately model human experimental results.** What is evident from human bargaining experiments is the presence of **irrationality** and **fairness issues** [1].

The purpose of our experiments is **to study the specific behavior** of humans (represented here by LLMs, as proxies for humans) in the bargaining problem. We expect that, in certain scenarios, **retaliatory strategies** might emerge, similar to phenomena observed in human bargaining experiments. If so, this implies that **human receivers do not fully behave according to the canonical analysis in classical BP**, especially when players may repeatedly interact.

In Appendix B.2, we pointed out that the persuasion community is currently concerned about long-term interactions. We also believe that some current work essentially designs algorithms to modify players’ bargaining power. For example, line 826 mentions [2] that the sender can commit to stopping communication if the receiver ignores their advice.

We first conducted bargaining experiments using LLMs, finding that the results align with observations from human experiments [1] and are intuitive. This indicates that LLMs can both comprehend and act as proxies for humans. For persuasion experiments, we only need to demonstrate that the receiver's behavior reflects retaliatory considerations like in the bargaining experiments. We also provided a primitive analysis of the impact of long-term interactions.

**Reference**

1. Lin, Po-Hsuan, et al. "Evidence of general economic principles of bargaining and trade from 2,000 classroom experiments." _Nature Human Behaviour_ 4.9 (2020): 917-927.
2. Bernasconi, Martino, et al. "Persuading farsighted receivers in mdps: the power of honesty." _Advances in Neural Information Processing Systems_ 36 (2024).

---

### Author Response · Authors · 2024-11-18
**Public Response (2): Experiments (b)**

**Experimental Setup**

**Our experiments did not involve any commitment by the receiver.** Instead, they strictly adhered to the scenarios described in **Examples 4.1–4.4**:

- Bargaining game: The proposer makes a reward assignment, which the responder then reviews. The responder chooses whether to accept or reject the offer. If accepted, the reward is divided according to the assignment. If rejected, the disagreement point is used for the division.
- Long-term bargaining game: This follows the standard Rubinstein alternating-offer model. If the responder rejects the current reward assignment, they then propose a new reward assignment (roles switch), and the process continues until an agreement is reached or a timeout occurs.
- Bayesian persuasion: The sender commits to a signaling scheme, and the receiver then decides their action rule. We have argued that having the receiver "commit" in advance to a Bayesian decision rule is inappropriate in canonical analysis (discussed in Public Response (1)). Therefore, we focus on pre-consensus scenarios, allowing the LLM to generate its own action rule. Additionally, the receiver evaluates their satisfaction with the outcome and explains their motivations. This enables us to observe whether they exhibit retaliatory behavior or fairness considerations.
- Long-term Bayesian persuasion: The sender and receiver roles are fixed. The sender always commits to a signaling scheme, while the receiver determines their action rule and states whether they are satisfied. This setup involves repeated games. If the receiver is dissatisfied, they default to the disagreement point's reward assignment and move to the next round. Once an agreement is reached, it is assumed that this agreement persists for all future rounds, terminating the process. Players’ strategies are history-dependent, and they are aware of the interaction history when making decisions.
- The receiver’s action rule can be determined after sampling a state, rather than beforehand. Therefore, this is not a prior commitment and does not alter the Bayesian persuasion procedure. For instance, after the sender commits to a signaling scheme, a student is sampled from the prior distribution, and the receiver calculates the mentioned considerations to decide their action rule. From this perspective, we are studying Bayesian persuasion itself, not a variant.

Our code and all experimental records were uploaded to the supplementary materials at the time of submission.

**Supplementary Experiments**

- We originally assumed that the sender’s commitment must require a long-term game. However, after further review, we found alternative justifications [1], such as the transparency of grading mechanisms in schools and the public enforcement of traffic laws. Thus, we will supplement our experiments with a one-shot Bayesian persuasion setting (the $\alpha$ setting).
- The $\gamma$ settings for bargaining and persuasion are not equivalent. In bargaining, players alternate roles in making offers, whereas this is not the case in persuasion. We will add an equivalent bargaining experiment that matches the persuasion setting.

**Reference**

 1. Kamenica, Emir. "Bayesian persuasion and information design." *Annual Review of Economics* 11.1 (2019): 249-272.

---

### Meta-Review · Area_Chair_NXZs · 2024-12-09

**Metareview:**

The paper argues that Bayesian persuasion can be reframed as a bargaining process when receivers can commit to strategies, countering the sender's informational advantage. The authors conduct theoretical analysis and simulations using LLM agents to illustrate how bargaining dynamics emerge naturally in persuasion tasks. The key claims include that Bayesian persuasion with receiver commitment resembles bargaining games and that receivers can employ anti-exploitation strategies when aware of the game structure.

The reviewers acknowledged the novel perspective and the effort to connect Bayesian persuasion and bargaining games. However, the main concerns are surrounding whether the claim of the paper is accurate or overstated.  There were additional concerns about the clarity of experimental design and the justification for using LLMs as proxies for human behavior. Overall, I tend to agree with the reviewers that the results of the paper might not be what it claims to be (at least not as strongly as suggested by the title and framing of the paper). While it could be the case that the reviewers and I misunderstood, it does suggest the paper would benefit from revisions to more clearly convey its contributions and/or reframing the paper.

**Additional Comments On Reviewer Discussion:**

The discussion primarily revolves around the framing of the paper's claims. While the gap between the authors and reviewers has narrowed, disagreements remain. After reading the discussion, I tend to agree more with the reviewers.  I want to note that the paper
does present interesting ideas and contributions, but it could benefit from more careful framing and positioning.

---

### Decision · Program_Chairs · 2025-01-22

Reject